# Multiscale fractal dimension analysis of a reduced order model of coupled ocean-atmosphere dynamics

Tommaso Alberti[1], Reik V. Donner[2, 3], and Stéphane Vannitsem[4]

[1]INAF-IAPS, via del Fosso del Cavaliere 100, 00133 Rome, Italy
[2]Department of Water, Environment, Construction and Safety, Magdeburg–Stendal University of Applied Sciences, Breitscheidstraße 2, 39114 Magdeburg, Germany
[3]Research Department IV – Complexity Science and Research Department I – Earth System Analysis, Potsdam Institute for Climate Impact Research (PIK) – Member of the Leibniz Association, Telegrafenberg A31, 14473 Potsdam, Germany
[4]Royal Meteorological Institute of Belgium, Bruxelles, Belgium

**Correspondence:** Tommaso Alberti (tommaso.alberti@inaf.it)

**Abstract.** Atmosphere and ocean dynamics display many complex features and are characterized by a wide variety of processes and couplings across different timescales. Here we demonstrate the application of Multivariate Empirical Mode Decomposition (MEMD) to investigate the multivariate and multiscale properties of a reduced order model of the ocean-atmosphere coupled dynamics. MEMD provides a decomposition of the original multivariate time series into a series of oscillating patterns with time-dependent amplitude and phase by exploiting the local features of the data and without any a priori assumptions on the decomposition basis. Moreover, each oscillating pattern, usually named Multivariate Intrinsic Mode Function (MIMF), represents a local source of information that can be used to explore the behavior of fractal features at different scales by defining a sort of multiscale/multivariate generalized fractal dimensions. With these two complementary approaches, we show that the ocean-atmosphere dynamics presents a rich variety of features, with different multifractal properties for the ocean and the atmosphere at different timescales. For weak ocean–atmosphere coupling, the resulting dimensions of the two model components are very different, while for strong coupling for which coupled modes develop, the scaling properties are more similar especially at longer time scales. The latter result reflects the presence of a coherent coupled dynamics. Finally, we also compare our model results with those obtained from reanalysis data demonstrating that the latter exhibit a similar qualitative behavior in terms of multiscale dimensions and the existence of a scale-dependency of the statistics of the phase-space density of points for different regions, which is related to the different drivers and processes occurring at different timescales in the coupled atmosphere-ocean system. Our approach can therefore be used to diagnose the strength of coupling in real applications.

## 1 Introduction

The atmosphere and the ocean form a complex system whose dynamical variability extends over a wide range of spatial and temporal scales (Liu, 2012; Xue et al., 2020). As an example, the tropical regions are markedly characterized by inter-/multi-

annual processes like the El Niño–Southern Oscillation (ENSO) (Neelin et al., 1994; Meehl et al., 2003), while the North Atlantic Oscillation (NAO) affects extra-tropical Northern hemispheric regions at seasonal and decadal timescales (Ambaum et al., 2001). The sources of these processes have been widely investigated by means of multiple data analysis methods and various types of modelling (e.g., Philander, 1990; Czaja and Frankignoul, 2002; Van der Avoird et al., 2002; Mosedale et al., 2006; Kravtsov et al., 2007; Feliks et al., 2011; Liu, 2012; L'Hévéder et al., 2014; Farneti, 2017; Vannitsem and Ghil, 2017; Wang, 2019; Xue et al., 2020, and references therein), highlighting how the atmospheric low-frequency variability (LFV) is related to the ocean. The latter develops thanks to the interaction with the ocean mixed layer (OML) driven by a mixing process due to the development of an instability within the water column (Czaja and Frankignoul, 2002; D'Andrea et al., 2005; Wunsch and Ferrari, 2004; Gastineau et al., 2012) that also shows a strong seasonal variability. The relation between the OML and the LFV can be investigated from a dynamical system point of view by developing suitable reduced order ocean-atmosphere models dealing with the modelling of the coupling between the atmosphere and the underlying surface layer of the ocean. Recently, by means of a 36-variable model displaying marked LFV Vannitsem et al. (2015) demonstrated that the LFV in the atmosphere could be a natural outcome of the ocean-atmosphere coupling. Other sources could be invoked to explain and to contribute to the development of LFV in the atmosphere, such as the long-range system memory as a consequence of the heat storage mechanism of the land-ocean-atmosphere system (e.g., Lovejoy, 2021; Lovejoy et al., 2021), the internal dynamics of the atmosphere itself (e.g., Legras and Ghil, 1985), or even the interaction between the tropical and extratropical regions (e.g., Alexander et al., 2002; Vannitsem et al., 2021), just to quote a few.

The current work presents an investigation on how a recently introduced concept of multiscale generalized fractal dimensions can be used to analyze the statistics of attractors in coupled ocean-atmosphere systems (Alberti et al., 2020a). This demonstration is done by means of the reduced order model developed in Vannitsem et al. (2015). Indeed, the dynamical properties of physical systems can be related to their support fractal dimension as well as its singularities by means of different established concepts like the box-counting dimension (e.g., Steinhaus, 1954; Mandelbrot, 1967; Ott, 2002), generalized correlation integrals (Grassberger, 1983; Hentschel and Procaccia, 1983; Pawelzik and Schuster, 1987), the pointwise dimension method (Farmer et al., 1983; Donner et al., 2011), and related characteristics (Badii and Politi, 1984; Primavera and Florio, 2020). These methods are based on partitioning the phase-space into hypercubes of size $\ell$ to define a suitable invariant measure through the filling probability of the $i-$th hypercube by $N_k$ points as $p_k = N_k/N$, with $N$ being the total number of points. With $M(\ell)$ denoting the number of filled hypercubes, we can define some useful dynamical invariants such as the box-counting (or capacity or simply fractal) dimension

$$D_0 \doteq -\lim_{\ell \to 0} \lim_{N \to \infty} \frac{\log M(\ell)}{\log \ell}, \tag{1}$$

the information dimension

$$D_1 \doteq \lim_{\ell \to 0} \lim_{N \to \infty} \frac{\sum_{k=1}^{M(\ell)} p_k \log p_k}{\log \ell}, \tag{2}$$

and the correlation dimension

$$D_2 \doteq \lim_{\ell \to 0} \lim_{N \to \infty} \frac{\frac{1}{N^2} \sum_{i \neq j} \Theta \left( \ell - |x_i - x_j| \right)}{\log \ell}, \tag{3}$$

with $\Theta(\cdots)$ being the Heaviside function. More specifically, $D_0$ is a measure of the sparseness of the phase-space by the studied system's dynamics, $D_1$ is an information measure giving us a measure of the information gained on the phase-space with a given accuracy, while $D_2$ is a measure of correlations, i.e., mutual dependence, between phase-space points. All these fractal dimension measures, as well as their higher order extensions $D_q$ measuring $q-$th order correlations between points in the phase-space, have been used to characterize the statistics of the phase-space scaling of a given system (Hentschel and Procaccia, 1983). More details on the estimation of generalized fractal dimensions are provided in the Supplementary Information. However, the above concepts only give us a global view on the phase-space system's properties, without exploring how these evolve at different scales in the real space (Alberti et al., 2020a). More recently, by means of a suitable combination between a state of the art time series decomposition method (the Empirical Mode Decomposition) and the concept of generalized fractal dimensions, Alberti et al. (2020a) introduced a multiscale approach to deal with the investigation of the evolution of the statistics of the phase-space scaling in dynamical systems.

Here, we extend for the first time the concept of multiscale generalized fractal dimensions in a multivariate framework by means of the Multivariate Empirical Mode Decomposition (MEMD), allowing us to investigate the multiscale and multivariate properties of a reduced order model of the ocean-atmosphere coupled dynamics. By using the oscillating patterns forming the decomposition basis of the MEMD algorithm, usually named Multivariate Intrinsic Mode Functions (MIMFs), we define the new concept of multiscale/multivariate generalized fractal dimensions. The MEMD results allow us to capture the essential dynamics of the phase-space trajectory that can be used for reconstructing the skeleton of the phase-space dynamics, while the evaluation of the fractal dimensions at different timescales provides a quantitative characterization of the intrinsic complexity of oscillating patterns that can be related to the attractor properties. Our results also allow for associating the statistics of the phase-space scaling to the dynamical regimes at different timescales of the coupled ocean–atmosphere system. Finally, our findings for the reduced order model well reconcile with corresponding results for reanalysis data, thus supporting and encouraging the use of reduced order models for investigating the essential aspects of the coupled ocean–atmosphere system in terms of the statistics of the phase-space scaling.

## 2    The reduced order ocean-atmosphere model

Reduced order coupled ocean-atmosphere models are key tools in the hierarchy of climate models, allowing for an extensive analysis of the features of the coupled dynamics that would otherwise be impossible to evaluate (Lorenz, 1984; Nese and Dutton, 1993; Roebber, 1995; Jin, 1996; Timmermann et al., 2003; Van Veen, 2003; De Cruz et al., 2016; Vannitsem, 2017). These models allow for obtaining key insights into the role of coupling for the development of LFV in the atmosphere associated with the presence of the ocean.

Recently, dynamical analysis has been conducted by means of the development of a suitable reduced order model of the coupled ocean-atmosphere system. This model has been developed starting from the quasi-geostrophic equations describing the interaction between a two-layer atmosphere and a one-layer ocean over an infinitely deep quiescent ocean layer (Vannitsem et al., 2015; Vannitsem, 2015; De Cruz et al., 2016; Vannitsem, 2017; De Cruz et al., 2018). The ocean flow passively advects

the temperature within the ocean, while momentum, radiative, and heat transfer mechanisms realize the coupling between the atmosphere and the ocean. By expanding the solutions of these equations into Fourier series, by truncating them at low wavenumbers, and by projecting onto the Fourier modes retained, a set of ordinary differential equations is derived. The fields are defined over a rectangular domain with $0 \leq x \leq 2\pi L/n$ and $0 \leq y \leq \pi L$, with $n$ denoting the aspect ratio between the meridional and the zonal extents of the domain and $L$ the characteristic spatial scale. Moreover, periodic boundaries along the zonal direction and free-slip along the meridional direction are chosen for the atmosphere, while a closed basin with no flux through the boundaries is imposed for the ocean.

In the reduced order coupled model version proposed in Vannitsem et al. (2015), a long-periodic attracting orbit combining atmospheric and oceanic variables emerges from a Hopf bifurcation for large values of the meridional gradient of radiative input and frictional coupling. Beyond a certain value of the meridional gradient for the radiative input, a chaotic behavior appears, which is still dominated by LFV on decadal and multi-decadal time-scales.

Here we use the original version of the model (Vannitsem et al., 2015) where the four relevant fields, i.e., the barotropic and baroclinic atmospheric streamfunctions, the ocean streamfunction and the ocean temperature, are given by $\psi_a = \sum_{i=1}^{10} \psi_{a,i} F_i$, $\theta_a = \sum_{i=1}^{10} \theta_{a,i} F_i$, $\Psi_o = \sum_{i=1}^{8} \Psi_{o,i} \phi_i$ and $T_o = \sum_{i=1}^{8} T_{o,i} \phi_i$, where $F_i$ and $\phi_i$ are simplified notations for the sets of modes used, compatible with the boundary conditions of both the atmosphere and the ocean. The parameter values used are the ones given in Figs. 8 and 9 of Vannitsem (2017). Depending on the choice of the surface friction coefficient $C$, different solutions are found with a highly chaotic dynamics without marked LFV in the atmosphere for small values of $C$, but a more moderately chaotic dynamics with stronger LFV in both the ocean and the atmosphere (related to the development of a coupled mode) for larger values of $C$.

## 3   Methods

Traditional multivariate and/or spatiotemporal data analysis methods are commonly based on fixing an orthogonal decomposition basis, satisfying certain mathematical properties such as linearity and/or stationarity (Chatfield, 2016). However, these conditions are not usually met when real-world geophysical data are analyzed, which calls for more adaptive methods (Huang et al., 1998). Indeed, adaptive methods can be helpful for overcoming some limitations of fixed-basis methods, implicitly assuming that a given natural phenomenon or a superposition of physical processes can be represented in terms of a priori defined mathematical functions like sine and/or cosine or some other kinds of wave functions (Chatfield, 2016). Since this cannot be assured, adaptive methods (as the MEMD) could be more suitable for reducing some mathematical assumptions and a priori constraints (Huang et al., 1998; Huang and Wu, 2008; Rehman and Mandic, 2010). Moreover, geophysical data are usually also characterized by scale-invariant features over a wide range of scales with different complexity and show a scale-dependent behavior due to several factors like forcings, coupling, intrinsic variability, and so on (e.g., Lovejoy and Schertzer, 2013; Franzke et al., 2020). For the above reasons, in this work we put forward a novel approach based on combining two different data analysis methods for investigating the multiscale fractal behavior of the coupled ocean-atmosphere system: Multivariate Empirical Mode Decomposition (MEMD; Rehman and Mandic, 2010) and generalized fractal dimensions (Hentschel

and Procaccia, 1983). One of the main advantages of combining the MEMD with generalized fractal dimensions instead of classical approaches deals with the limited number of intrinsic components that can be also visually inspected. Indeed, if we, for example, use Fourier decomposition we will have a large number of (harmonic) oscillating components at different fixed frequencies that should be summed up for exploiting our proposed procedure. Furthermore, if we, for example, use wavelets we will deal with some a priori assumptions on the decomposition basis onto which we are projecting our data that could produce misleading results in our procedure of evaluating fractal measures on a priori fixed scales. Another advantage is that MEMD allows to preserve some intrinsic properties of signals related to the nonlinear and/or non-stationary nature of processes they are associated with, since the decomposition is based on the local characteristic scale of the data in deriving intrinsic components with time-dependent amplitudes and phases (Huang et al., 1998; Huang and Wu, 2008; Rehman and Mandic, 2010). However, we do not question the appropriateness of conventional analysis techniques, but rather acknowledge that other approaches can provide a new perspective on what we can learn from the respective system under study (Alberti et al., 2020a).

## 3.1 Multivariate Empirical Mode Decomposition (MEMD)

The Multivariate Empirical Mode Decomposition (MEMD) is the "natural" multivariate extension of the univariate Empirical Mode Decomposition (EMD) (Huang et al., 1998; Rehman and Mandic, 2010). MEMD directly works on the data domain, instead of defining a conjugate space as for Fourier or Wavelet transforms, with the aim of being as adaptive as possible to minimize mathematical assumptions and definitions (Huang et al., 1998) in extracting embedded structures in the form of so-called Multivariate Intrinsic Mode Functions (MIMFs) (Rehman and Mandic, 2010). Each MIMF is an oscillatory pattern of the multivariate coordinates having the same number (or differing at most by one) of local extremes and zero crossings, and whose upper and lower envelopes are symmetric (Huang et al., 1998; Rehman and Mandic, 2010). MIMFs are derived through the sifting process (Huang et al., 1998). This process is easily realized for univariate signals (Huang et al., 1998), while needs to be carefully implemented for multivariate processes (Rehman and Mandic, 2010), since it is based on the cubic spline interpolation of local extremes that cannot easily be defined on a $k$-dimensional space (Rehman and Mandic, 2010). Rehman and Mandic (2010) proposed an alternative definition of local extremes for multivariate signals by considering the $k$-variate data as composed by $k$-dimensional signals projected onto appropriate directions in this $k$-dimensional space. This allows us to perform cubic spline interpolation in each direction, with the suitable directions chosen by means of a combination of a quasi-Monte Carlo-based low-discrepancy sequences and a uniform angular sampling method (Rehman and Mandic, 2010). These allow to provide a more uniform set of direction vectors over which to compute the local mean of envelopes, without introducing any smoother dynamics in the data, via the following procedure:

1. given a $k$-dimensional space we need to find the direction vectors by considering that these reduce to points in a ($k$-1)-dimensional space;

2. the simplest choice is to employ uniform angular sampling on a $k$-dimensional hypersphere but this will lead to a non-uniform filling of the $k$-dimensional space (a higher density of points would be observed near the poles);

3. a quasi-Monte Carlo method is used to provide a more uniform distribution of direction vectors;

4. once the direction vectors are chosen, the signal is projected onto these vectors, the extrema of the resulting projected signals are evaluated and interpolated component-wise to yield multidimensional envelopes that are then averaged to obtain the multivariate mean.

This means that the quasi-Monte Carlo method is needed only for selecting a uniform sampling of direction vectors, thus to avoid implicitly preferred directions that could be more dominant with respect to the others, which could introduce a source of errors in evaluating signal projections (Rehman and Mandic, 2010).

Having now defined the procedure needed to compute envelopes over each direction, the main steps of the sifting process acting on a $k$-variate signal $\mathbf{s}(t) = [s_1(t), s_2(t), \ldots, s_k(t)]$ can be summarized as below:

1. identify local extremes (i.e., data points where abrupt changes in the local tendency of the series under study are observed);

2. interpolate local extremes separately by cubic splines (i.e., produce continuous functions with smaller error than other polynomial interpolation);

3. derive the upper and lower envelopes $\mathbf{u}(t)$ and $\mathbf{l}(t)$, respectively;

4. derive the mean envelope $\mathbf{m}(t)$ as $\mathbf{m}(t) = \frac{\mathbf{u}(t) + \mathbf{l}(t)}{2}$;

5. evaluate the resulting candidate MIMF as $\mathbf{h}(t) = \mathbf{s}(t) - \mathbf{m}(t)$.

The previous steps are iteratively repeated until the obtained candidate MIMF $\mathbf{h}(t)$ can be identified as a Multivariate Intrinsic Mode Function (also called multivariate empirical mode) (Huang et al., 1998; Rehman and Mandic, 2010), while the full sifting process ends when no more MIMFs $\mathbf{c}_j(t)$ can be filtered out from the data. Hence, we can write

$$\mathbf{s}(t) = \sum_{j=1}^{N_j} \mathbf{c}_j(t) + \mathbf{r}(t). \tag{4}$$

In this way a multivariate signal is decomposed into $N_j$ $k$-dimensional functions, each containing the same frequency distribution, e.g., into a set of $k$-dimensional embedded oscillating patterns $\mathbf{c}_j(t)$ which form the multivariate decomposition basis, plus a multivariate residue $\mathbf{r}(t)$.

For each MIMF we can define a $k^\star-$variate mean timescale as

$$\tau_{j,k^\star} = \frac{1}{T} \int_0^T t' c_{j,k^\star}(t') dt', \tag{5}$$

representing the typical oscillation scale of the $j-$th mode for the $k^\star$-th univariate component $c_{j,k^\star}$ extracted from the multivariate signal $s_{k^\star}(t)$ for $k^\star \in [1, k]$. Similarly, by ensemble averaging over the $k$-dimensional space we can introduce the concept of a multivariate mean timescale as

$$\tau_j = \frac{1}{T} \int_0^T t' \langle \mathbf{c}_j(t') \rangle_k dt', \tag{6}$$

with $\langle\cdots\rangle_k$ denoting an ensemble average over the $k$-dimensional space. Thus, the $k^\star-$variate timescale $\tau_{j,k^\star}$ is evaluated for each mode and for each $k^\star-$dimensional data, while the multivariate mean timescale $\tau_j$ is the mean over all $k^\star \in [1,k]$. Moreover, as for univariate EMD (Huang et al., 1998), we can introduce the concepts of instantaneous amplitudes $\mathbf{a}_j(t)$ and phases $\phi_j(t)$ of each MEMD mode via the Hilbert Transform along the different directions of the $k$-dimensional space. The instantaneous energy content is then derived as $\mathbf{E}_j(t) = \mathbf{a}_j(t)^2$. Thereby, we can characterize the spectral content by introducing an alternative yet equivalent definition of the power spectral density (PSD) as

$$S(\tau) = \frac{1}{T^2}\int_0^T \langle \mathbf{E}_j(t')\rangle_k dt' \cdot \int_0^T t'\langle \mathbf{c}_j(t')\rangle_k dt' \doteq \sigma^2(\tau) \cdot \tau, \tag{7}$$

with $\sigma^2(\tau)$ being the $k-$variate variance of MIMFs and $\tau$ the mean timescale defined as in Eq. (6). Moreover, from the instantaneous energy content $\mathbf{E}_j(t)$ the relative contribution $e_j$ can be derived as

$$e_j = \frac{\frac{1}{T}\int_0^T \langle \mathbf{E}_j(t')\rangle_k dt'}{\sum_{j=1}^{N_j} \frac{1}{T}\int_0^T \langle \mathbf{E}_j(t')\rangle_k dt'}. \tag{8}$$

Finally, as for the univariate decomposition (Huang et al., 1998), also the MIMFs are empirically and locally orthogonal with respect to each other, the decomposition basis is a complete set (Rehman and Mandic, 2010) and partial sums of Eq. (4) can be obtained (Alberti, 2018; Alberti et al., 2020b).

## 3.2 Multivariate and multiscale generalized fractal dimensions

The dynamics of complex systems is usually characterized by a multitude of scales whose dynamical features determine their collective behavior. Nevertheless, vast efforts have been made to determine collective properties of systems (e.g., Hentschel and Procaccia, 1983), instead of considering to measure scale-dependent features. Recently, Alberti et al. (2020a) introduced a new formalism allowing measuring information at different scales by combining a data-adaptive decomposition method and the classical concept of generalized fractal dimensions. The starting point is that a multivariate signal manifesting a multiscale behavior can be written as

$$\mathbf{s}(t) = \langle \mathbf{s}\rangle + \sum_\tau \delta\mathbf{s}_\tau(t) = \mathbf{s}_0 + \mathbf{s}_1(t), \tag{9}$$

with $\langle\cdots\rangle$ representing a steady-state average operation and $\delta$ indicating a fluctuation at scale $\tau$. For any given $\tau$ we can introduce a local natural probability measure $d\mu_\tau$ such that the probability $p_i$ of visiting the $i-$th hypercube $B_{\mathbf{s}^*,\tau}(\ell)$ of size $\ell$ centered at the point $\mathbf{s}^*$ on the considered ($d-$dimensional) phase-space of $\mathbf{s}_1(t)$ can be defined as

$$p_i \doteq \int_{\mathbf{s}_1 \in B_{\mathbf{s}^*,\tau}(\ell)} d\mu_\tau. \tag{10}$$

By defining a $q-$th order partition function

$$\Gamma_q(\mu_\tau, B_{\mathbf{s}^*,\tau}(\ell)) = \sum_i p_i^q = \int d\mu_\tau(s)\mu_\tau(B_{\mathbf{s}^*,\tau}(\ell))^q \tag{11}$$

and taking the limit $\ell \to 0$, the multiscale generalized fractal dimensions are derived as

$$D_{q,\tau} = \frac{1}{q-1} \lim_{\ell \to 0} \frac{\log \Gamma_q(\mu_\tau, B_{\mathbf{s}^*,\tau}(\ell))}{\log \ell}. \tag{12}$$

Here we identify the intrinsic oscillations by using the MEMD and then we investigate the phase-space properties at different scales by deriving the generalized dimensions (Alberti et al., 2020a). Summarizing:

1. we extract multiscale components from $\mathbf{s}(t)$ by using the MEMD;

2. we evaluate the intrinsic scale $\tau_j$ of each MIMF;

3. we evaluate reconstructions of modes by means of Eq. (4)

$$\sum_\tau \delta \mathbf{s}_\tau(t) \to F_{j^\star}(t) = \sum_{j=1}^{j^\star} \mathbf{c}_j(t) \tag{13}$$

with $j^\star = 1, \ldots, N_j$ (by construction, MIMFs are ordered from short to long scales, i.e., $\tau_j < \tau_{j'}$ if $j < j'$);

4. we evaluate the generalized dimensions $D_{q,\tau}$ from $F_{j^\star}(t)$ for each $j^\star$ (i.e., for each scale $\tau_{j^\star}$),;

5. we evaluate the singularities and singularity spectrum

$$\alpha_\tau = \frac{d}{dq}\left[(q-1)D_{q,\tau}\right] \tag{14}$$

$$f_\tau = f(\alpha_\tau) = q\alpha_\tau - \left[(q-1)D_{q,\tau}\right]. \tag{15}$$

From Eq. (13) we can inspect the local properties of fluctuations in terms of the geometry of the phase-space, thus providing a characterization of dynamical features of different regimes and disentangling the different dynamical components of (possibly) different origin.

Our proposed formalism provides a novel way to investigate how phase-space properties (geometry, correlations) change when dynamical components at different mean scales with different dynamics are considered. In other words, we can highlight the role of scale-dependent phenomena in defining the global properties of a system. Indeed, global measures proposed in the past (e.g., Grassberger, 1983; Hentschel and Procaccia, 1983) only allow us to investigate the statistics of the phase-space scaling properties of the whole system; conversely, our proposed approach allows us to investigate how the different scales contribute to the global properties of a system (Alberti et al., 2020a). Moreover, our framework also provides consistency with established measures for characterizing time series from an integral (not scale-resolved) perspective, since the scale-dependent measures we evaluate converge to the associated global measures as all scales are considered, i.e., when the full system dynamics, composed by all accessible scales, is reached (Hentschel and Procaccia, 1983). Within this framework, our approach is promising for investigating scale-dependent properties, as measured by fractal dimensions, of the system. Furthermore, since we are indeed interested in nonlinear variability characteristics at different time scales, employing perfectly linear and/or stationary (harmonic) functions as components would leave out any information on nonlinear dynamics. Moreover, simply

looking at the behavior of spectral densities would leave out any higher-order statistical properties, only focusing on the auto-correlation function (i.e., the second-order moment). By looking at the behavior of fractal dimensions we can explore how the different scales contribute to change the phase-space properties for higher-order statistics (i.e., for different values of $q$).

# 4  Results

## 4.1  Multivariate Empirical Mode Decomposition

Figure 1 reports the 3-D projection of the full system attractor in the subspace $(T_{o,2}, \Psi_{o,2}, \psi_{a,1})$ for two representative values of the friction coefficient $C$ (0.008 and 0.015 kg m$^{-2}$ s$^{-1}$ as indicated by red and black points, respectively). In the following, we will omit the physical units of this parameter for the sake of brevity. The considered subspace characterizes the dynamics of the system as represented by the dominant mode of the meridional temperature gradient in the ocean ($T_{o,2}$), by the double-gyre transport within the ocean ($\Psi_{o,2}$), and by the vertically averaged zonal flow within the atmosphere ($\psi_{a,1}$), respectively.

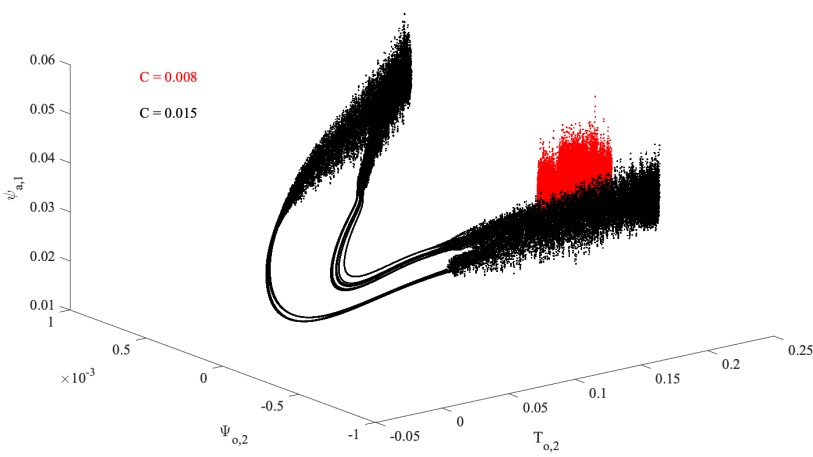

**Figure 1.** 3-D projection of the full system attractor in the subspace $(T_{o,2}, \Psi_{o,2}, \psi_{a,1})$ for $C = 0.008$ (red) and $C = 0.015$ (black), respectively.

The behavior of the system is clearly dependent on the friction coefficient, with both the location and the topology of the attractor changing as $C$ is increased from $0.008$ (red points in Fig. 1) to $0.015$ (black points in Fig. 1). This behavior has also been previously reported by Vannitsem et al. (2015) and Vannitsem (2015), indicating a drastic qualitative change of the nature of the dynamics at about $C = 0.011$ above which substantial LFV emerges (Vannitsem et al., 2015; Vannitsem, 2015, 2017). However, all model components are clearly characterized by multiscale variability, spanning a wide range of timescales that

can contribute to the dynamics in different ways, depending on the values of the friction coefficient and the intrinsic variability of the coupled ocean-atmosphere system.

Figure 2 displays the behavior of the spectral energy content $S(\tau)$ of the different MIMFs as a function of their mean timescales $\tau$ as in Eq. (7) for the full system (atmosphere+ocean) and for the two subsystems separately (i.e., the atmosphere and the ocean, respectively). First of all, it is important to underline that a different number of MIMFs has been identified

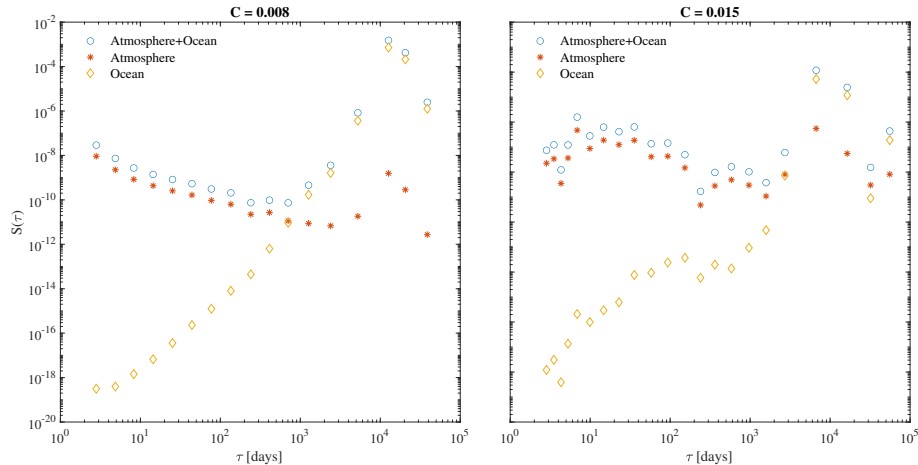

**Figure 2.** Spectral energy content $S(\tau)$ of the different MIMFs as a function of their mean timescales $\tau$ as in Eq. (7) for the full system (atmosphere+ocean, blue circles), only for the atmosphere (orange asterisks), and only for the ocean (yellow diamonds). Left and right panels refer to the two values of the friction coefficient, $C = 0.008$ and $C = 0.015$, respectively.

for the two different cases: $N_j = 17$ for $C = 0.008$ and $N_j = 22$ for $C = 0.015$. This underlines that the respective dynamical behavior of the system is different, being characterized by different sets of empirical modes and consequently by a different number of relevant timescales. Moreover, by keeping in mind that for pure noise the expected number of MIMFs is $\log_2 N$ with $N$ being the number of data points, both situations cannot be related with a purely stochastic dynamics. Indeed, in both
cases we have used $N = 10^5$ data points, thus the expected number of MIMFs is $N_j^{\text{noise}} = 16$ (Flandrin et al., 2004). However, an interesting feature is that for the lower $C$ value a number of MIMFs closer to that expected for noisy data is found, possibly related to the more irregular dynamics in this low friction coefficient case. Conversely, a marked departure from $N_j = 16$ is found for the higher $C$ case, corresponding to a more regular dynamics characterized by significant LFV.

Furthermore, from Fig. 2 it is easy to note that the behavior of $S(\tau)$ depends on both the friction coefficient $C$ and the dif-
ferent components of the model. For the full system (i.e., atmosphere+ocean) $S(\tau)$ decreases as $\tau$ increases for both values of $C$, while it is characterized by increasing spectral energy content at larger scales (i.e., at lower frequencies). By discriminating between the atmospheric and the oceanic contribution we are able to see that (as expected), the short-term variability of the full system can be attributed to the atmosphere, while the long-term one is a reflection of the ocean dynamics. Moreover, when $C$ increases we note an increase of the spectral energy content at all timescales, together with a flattening of the atmospheric
spectral behavior, while the ocean dynamics seems to preserve its spectral features. These behaviors can be related to the exis-

tence of multiscale variability of the full system that can be linked to the different components operating at different timescales and to the different dynamics of the system as the friction coefficient $C$ is changed.

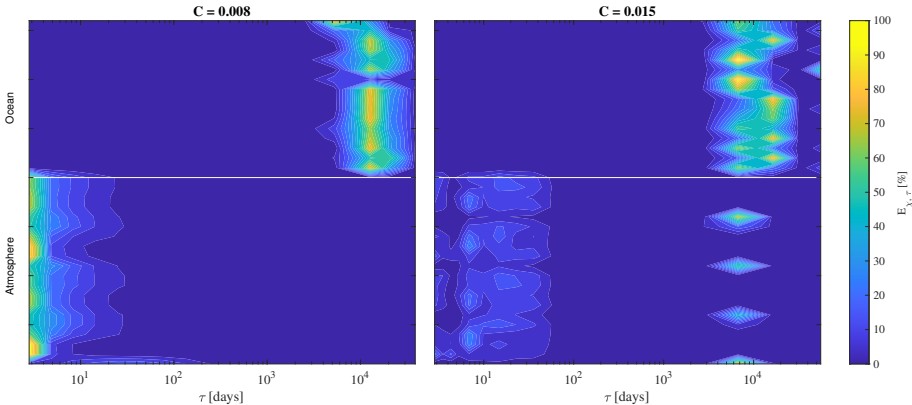

**Figure 3.** Relative contribution (in percentage) $E_{\chi,\tau}$ of each variable $\chi = \{\psi_{a,i}, \theta_{a,i}, \Psi_{o,i}, T_{o,i}\}$ in dependence on the mean timescale $\tau$. Left and right panels refer to the two values of the friction coefficient $C = 0.008$ and $C = 0.015$, respectively. The white line separates the atmospheric variables from the oceanic ones.

To further clarify the latter aspect, we evaluate the relative contribution (in percentage) $E_{\chi,\tau}$ of the different MIMFs (i.e., at different timescales $\tau$) for each variable $\chi = \{\psi_{a,i}, \theta_{a,i}, \Psi_{o,i}, T_{o,i}\}$ as reported in Fig. 3. It can be clearly noted that the oceanic
variability mainly contributes to the low-frequency dynamics ($E_{\chi,\tau} > 95\%$ for $\chi = \{\Psi_{o,i}, T_{o,i}\}$ and $\tau \gtrsim 10^4$ days), while the atmosphere is mainly characterized by short-term variability for $C = 0.008$ ($E_{\chi,\tau} > 95\%$ for $\chi = \{\psi_{a,i}, \theta_{a,i}\}$ and $\tau \lesssim 10$ days) and by both short- and long-term dynamics for $C = 0.015$. This points towards the $C$-dependent behavior of the atmospheric dynamics, with the ocean multiscale variability being less affected by changes in the values of the friction coefficient, and to the role of the ocean in developing LFV in the atmosphere as $C$ increases.
Thanks to the completeness property of the MEMD we can explore the dynamics of the system as reproduced by the most energetic empirical modes via partial sums of Eq. (4). By using the information coming from the energy percentage distribution across the different timescales for each variable $\chi$ we can provide MIMF reconstructions accounting for a certain percentage of energy with respect to the total spectral energy content. By ordering the empirical modes with decreasing relative contribution $e_j$ and summing up those contributing at least 95% of the total spectral content, we are able to investigate the 3-D projection
of the full system attractor onto the subspace $(T_{o,2}, \Psi_{o,2}, \psi_{a,1})$ and compare it with the projection obtained by considering all timescales (as in Figure 1). Thus, for each variable $\chi = \{\psi_{a,i}, \theta_{a,i}, \Psi_{o,i}, T_{o,i}\}$ we can define a reconstruction based on empirical modes, $R_{\chi,95\%}$, as

$$R_{\chi,95\%}(t) \doteq \sum_{j'|e_{j'} \geq 95\%} \mathbf{c}_{\chi,j'}(t) \tag{16}$$

with $\mathbf{c}_{\chi,j'}(t)$ being the $j'-$th multivariate empirical mode extracted via the MEMD of the variables $\chi$. The 3-D projection onto the subspace $(T_{o,2}, \Psi_{o,2}, \psi_{a,1})$ of $R_{\chi,95\%}$ is shown in Fig. 4, while Tab. 1 summarizes the mode indices $j'$ and corresponding $k^\star-$variate timescales $\tau_{j',k^\star}$ (see Eq. (5)) used for the reconstruction.

**Table 1.** Mode indices $j'$ and corresponding $k^\star-$variate timescales $\tau_{j',k^\star}$ (see Eq. (5)) used for the reconstruction based on empirical modes $R_{\chi,95\%}$.

| $C$ | $\chi$ | $j'$ | $\tau_{j',k^\star}$ [days] |
|---|---|---|---|
| | $\psi_{a,1}$ | 1, 2 | 3, 5 |
| 0.008 | $\Psi_{o,2}$ | 14, 15, 16 | 631, 1333, 2086 |
| | $T_{o,2}$ | 14, 15, 16 | 599, 1132, 1913 |
| | $\psi_{a,1}$ | 21 | 2690 |
| 0.015 | $\Psi_{o,2}$ | 19, 20, 21 | 829, 1469, 2449 |
| | $T_{o,2}$ | 19, 20, 21 | 735, 1506, 2598 |

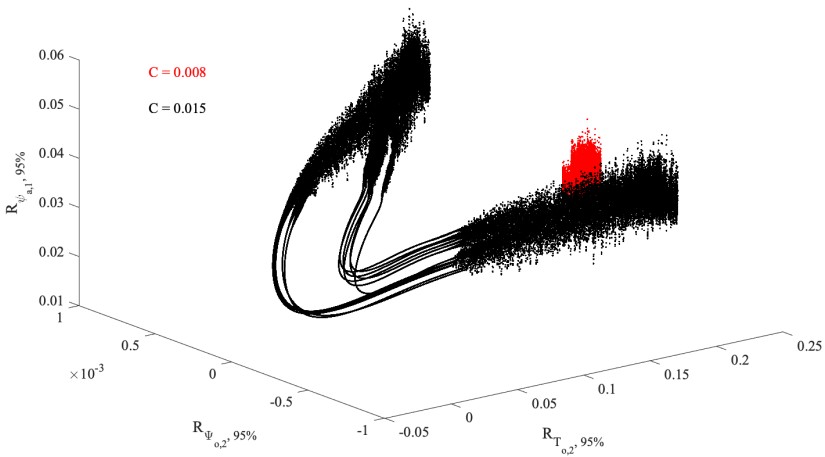

**Figure 4.** 3-D projection of the full system attractor in the subspace $(T_{o,2}, \Psi_{o,2}, \psi_{a,1})$ for $C = 0.008$ (red) and $C = 0.015$ (black), respectively, as obtained from reconstructions based on the multivariate empirical modes $R_{\chi,95\%}(t)$ accounting for 95% of the total variance of the model dynamics.

By comparing Figs. 1 and 4 it can be easily noted that the underlying structure of the 3-D projection of the full attractor is essentially the same, thus suggesting that the subspace statistics of the phase-space scaling information can be recovered by a subset of multivariate empirical modes. This underlines that the dynamics of the full system can be reproduced by only few relevant timescales without too much loss of information, thus reducing the complexity of the low order model itself. These

results appear relevant if put into the wider context of coupled ocean-atmosphere dynamics, allowing us to recover the main features by only considering the most relevant (in terms of energy) timescale dynamical components.

## 4.2 Multiscale generalized fractal dimensions

Under general conditions, the complexity of a dynamical system can be conveniently investigated by means of the nonlinear properties of its phase-space trajectory (e.g., its attractor or repellor in case of dissipative dynamics) (Ott, 2002). One of the most common ways to characterize the topology of an attractor is to compute its spectrum of generalized fractal dimensions, allowing us to statistically characterize important properties of the dynamics as reflected by its phase-space geometry, including its information content, complexity, and underlying fractal structure (Grassberger, 1983; Hentschel and Procaccia, 1983; Donner et al., 2011). However, classical approaches can only provide global information on the phase-space topology (Hentschel and Procaccia, 1983; Ott, 2002), while multiscale dynamical systems can be characterized by the statistics of the phase-space scaling changing as different real-space scales are considered (Alberti et al., 2020a). For this purpose, we investigate the statistics of the phase-space scaling of the coupled ocean-atmosphere model by evaluating the multiscale generalized fractal dimensions described in Section 3.2. Figures 5 and 6 report the behavior of the correlation dimension $D_2$ for both values of the friction coefficient and for three different cases: (a) for each MIMF individually ($D_2^j$), (b) for reconstructions of MIMFs ($D_{2,\tau}$), and (c) for reconstructions of MIMFs performed separately for each variable $\chi = \{\psi_{a,i}, \theta_{a,i}, \Psi_{o,i}, T_{o,i}\}$.

As expected, the multiscale correlation dimension for each MIMF decreases with increasing timescale, being representative of a more regular, less stochastic/chaotic, behavior of large-scale MIMFs as compared with the short-term ones (Alberti et al., 2020a). Particularly, when approaching the largest timescales, $D_{2,\tau} \to 1$ suggesting the existence of fixed-scale MIMFs, i.e., with the instantaneous frequencies being almost constant (as expected, e.g., Rehman and Mandic, 2010). Conversely, when the multiscale correlation dimensions are evaluated by summing up the different MIMFs, starting from the shortest up to the largest scale, a clearly scale-independent behavior of $D_{2,\tau}$ is highlighted for both values of the friction coefficient $C$. This suggests that the short-term variability mostly defines the correlations between pairs of points in the phase-space, thus setting the minimum number of variables needed to describe the dynamics of the system, i.e., its degrees of freedom. However, the role of $C$ clearly emerges in determining the values of $D_{2,\tau}$, being lower for the larger $C$ value. Indeed, $D_{2,\tau} \sim 8$ for $C = 0.008$, while $D_{2,\tau} \sim 1.5$ for $C = 0.015$. This reflects the different statistics of the attractor scaling of the full system associated with a different dynamical behavior of the model variables (Faranda et al., 2019), also suggesting a less chaotic nature of the system as $C$ increases, together with a reduced number of degrees of freedom. This points towards the possibility of recovering the main features of the model with a reduced number of variables and scales. However, the most interesting features emerge when the different variables of both atmosphere and ocean are separately investigated by means of the multiscale generalized fractal dimensions. It is indeed evident that a scale-independent behavior is found for the atmosphere for both values of $C$, while a scale-dependent behavior is observed for the ocean. The former can be easily related to the dominant role of the short-term variability for the atmosphere, while the latter is a reflection of the long-term dynamics of the ocean. Moreover, it is also particularly interesting to note that higher (lower) $D_{2,\tau}$ values are found for the atmosphere with respect to the ocean for $C = 0.008$ ($C = 0.015$). This reflects the role of the ocean in developing LFV in the atmosphere as $C$ increases, although the

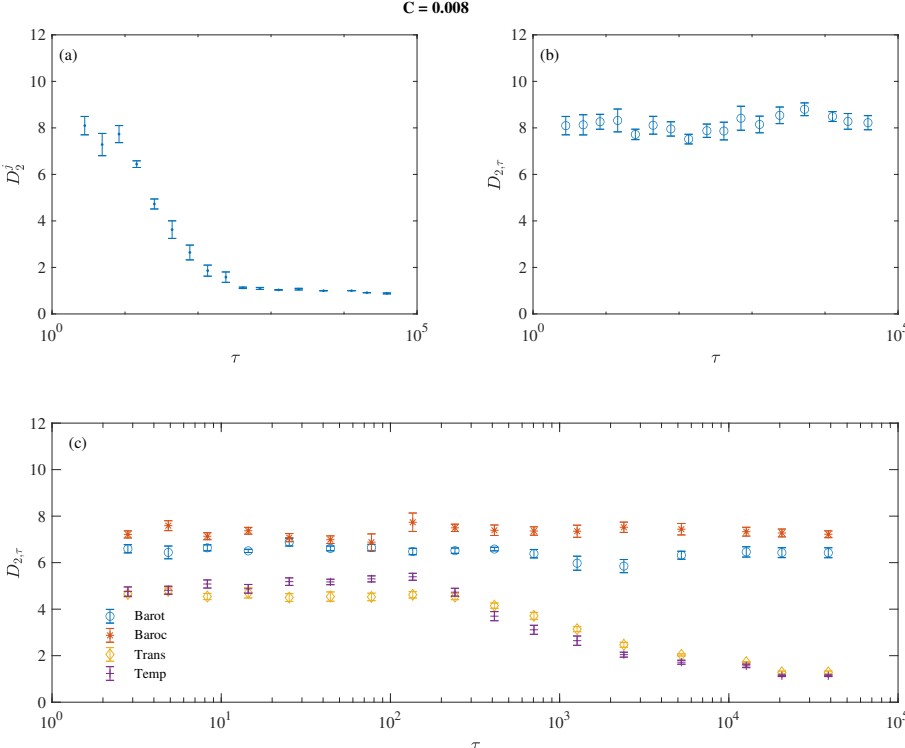

**Figure 5.** Multiscale correlation dimension $D_{2,\tau}$ for $C = 0.008$ at different timescales $\tau_j$ for different cases: (a) for each MIMF individually ($D_2^j$), (b) for reconstructions of MIMFs as in Eq. (12) ($D_{2,\tau}$), and (c) for reconstructions of MIMFs separately for each variable (barotropic modes - blue circles, baroclinic modes - orange asterisks, transport modes - yellow diamonds, and temperature modes - violet symbol). Each panel also shows the 95% confidence intervals as error bars.

complexity of the full system seems to be determined by the atmosphere for both $C$ values, being indeed characterized by a scale-independent behavior of $D_{2,\tau}$.

The described findings are not only valid for the multiscale correlation dimension $D_{2,\tau}$ but are also observed for both the multiscale capacity dimension $D_{0,\tau}$ and the multiscale information dimension $D_{1,\tau}$ as reported in Figs. 7 and 8, together with the multiscale correlation dimension $D_{2,\tau}$, for both values of $C$.

Our formalism reveals the expected property that for $q < q'$, $D_{q,\tau} > D_{q',\tau} \,\forall \tau$ (Hentschel and Procaccia, 1983; Alberti et al., 2020a). Moreover, when evaluating the multiscale generalized fractal dimensions for each MIMF separately (e.g., Figs. 7(a) and 8(a)) a decreasing value for $D_q^j$ is found as $\tau$ increases, with all $D_q^j$ converging towards the same value of 1 at large timescales. As for $D_2^j$ this behavior can be easily interpreted in terms of more chaotic vs. more regular MIMFs when moving from short to large scales. This indeed reflects the existence of large-scale MIMFs that are characterized by a linear phase, i.e.,

a constant timescale (e.g., Rehman and Mandic, 2010). Thus, this is a trivial result. Conversely, when the $D_{q,\tau}$ are evaluated for reconstructions based on MIMFs a scale-independent behavior is found for the full system for both values of $C$ (e.g., Figs. 7(b)

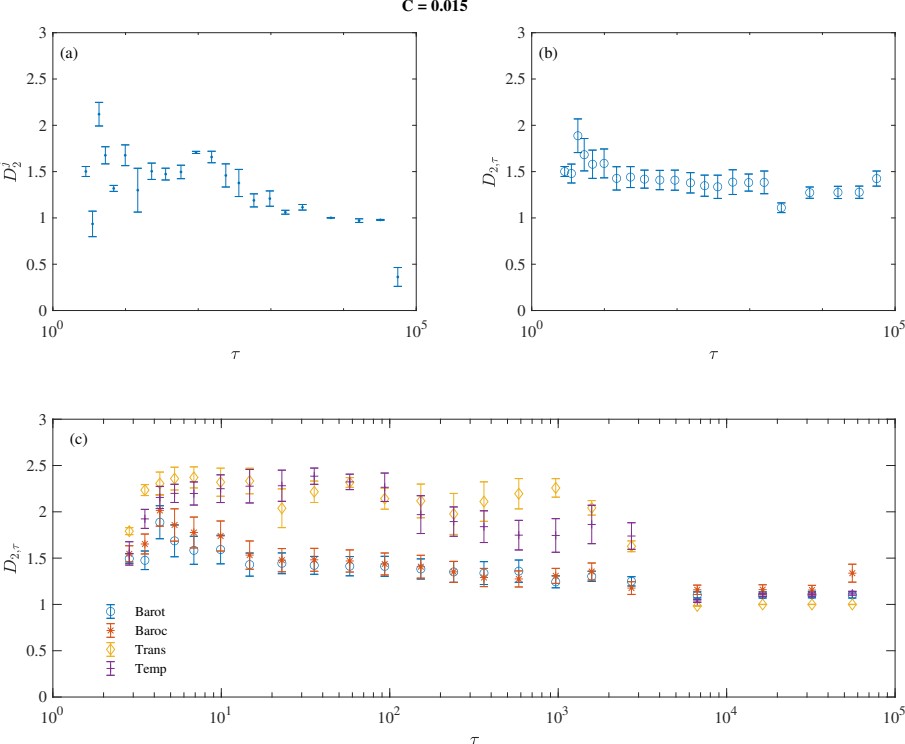

**Figure 6.** Same as in Fig. 5, but for $C = 0.015$.

and 8(b)). However, the key role of the friction coefficient clearly emerges by looking at the larger values of $D_{q,\tau}$ for $C = 0.008$ with respect to the lower values found for $C = 0.015$. This clearly indicates the existence of a completely different dynamics between the two values of $C$, where the coupled ocean-atmosphere dynamics can be interpreted as a higher-dimensional

chaotic system for reduced ocean-atmosphere coupling (i.e., $C = 0.008$) as opposed to a lower-dimensional one for a strong ocean-atmosphere coupling (i.e., $C = 0.015$). Although $C$ acts as a control parameter for the dimensionality of the system, it is not able to change the underlying fractal nature of the full system. Indeed, for both $C$ values we clearly observe different $D_{q,\tau}$ for different $q$, thus suggesting the existence of a multifractal nature of the ocean-atmosphere dynamics at all timescales. Furthermore, by separately looking at the two subsystems (i.e., the ocean and the atmosphere) a completely different behavior

emerges (e.g., Figs. 7(c)-(f) and 8(c)-(f)). In this case, the atmospheric variables are characterized by scale-independent $D_{q,\tau}$, being representative of a high-dimensional system whose prime dynamics occurs at short timescales and with little effects of large-scale processes on the collective dynamics of the atmosphere. By contrast, a clearly scale-dependent behavior is found for the oceanic variables, with the multiscale generalized dimensions decreasing at larger timescales, reflecting the effects of large-scale dynamics dominating with respect to the short-term one for the ocean variability. Again the friction coefficient $C$

controls the values of $D_{q,\tau}$, decreasing as $C$ increases, while both the atmosphere and the ocean are clearly characterized by multifractal features at all timescales.

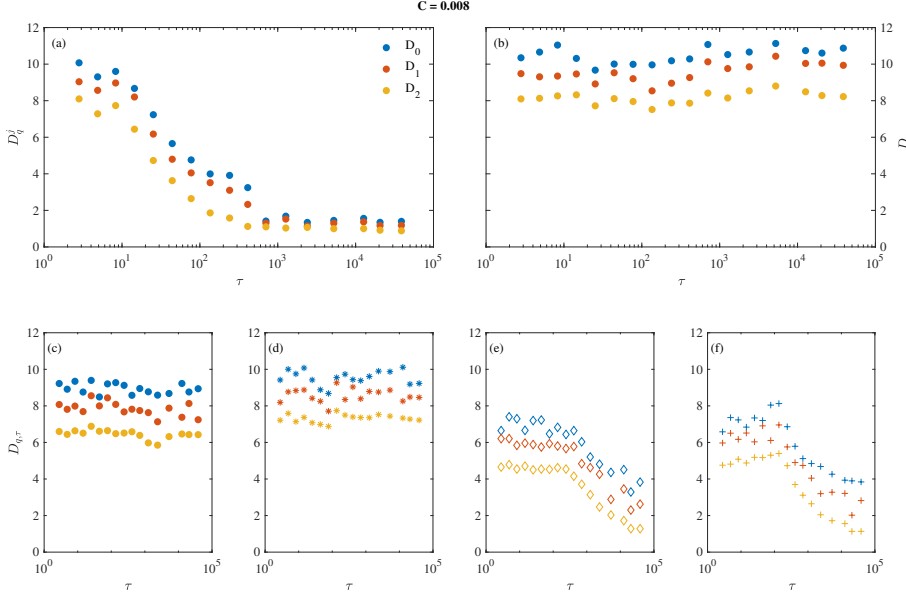

**Figure 7.** Multiscale capacity dimension $D_{0,\tau}$, multiscale information dimension $D_{1,\tau}$, and multiscale correlation dimension $D_{2,\tau}$ for $C = 0.008$ at different timescales $\tau_j$ for different cases: (a) for each MIMF individually ($D_q^j$), (b) for reconstructions of MIMFs as in Eq. (12) ($D_{q,\tau}$), and (c)-(f) for reconstructions of MIMFs separately for each variable (barotropic modes - (c), baroclinic modes - (d), transport modes - (e), and temperature modes - (f)).

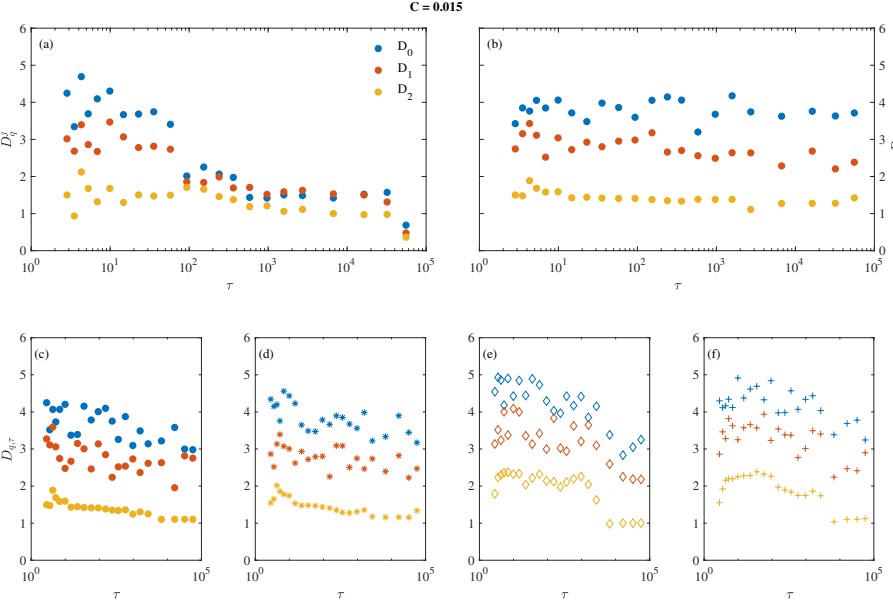

**Figure 8.** Same as in Fig. 7, but for $C = 0.015$.

By estimating the Lyapunov spectra (cf. Fig. S11 in Supplementary Information) separately for the ocean and the atmosphere we obtained that for $C = 0.008$ the instability is large for the atmosphere with a Lyapunov dimension $D_L \sim 10$, while for $C = 0.015$ the instability is weaker for the atmosphere, and the Lyapunov dimension is slightly larger than 4. Following the

Kaplan-Yorke conjecture (Kaplan and Yorke, 1979), the Lyapunov dimension can be used as a proxy of $D_0$. Hence, our results are clearly consistent with the dimension estimates for the atmosphere. For the ocean, however, there seems to be a less good agreement, with $D_L \approx 2$ while we found that $D_{0,\tau} \approx 4$. This quantitative disagreement could be related to the fact that the ocean can be viewed as a relatively stable system perturbed by high-frequency "noise" provided the atmosphere. Deeper investigations will be devoted to clarify this point in future research.

As a further step, we evaluate the full spectrum of generalized fractal dimensions for each MIMF by considering a wide range of statistical moments $q$. As suggested in Lovejoy and Schertzer (2013) the range of significant moments can be evaluated by means of the tail of the cumulative distribution function of the data. Indeed, the effect of sample size and its implications for spurious scaling may be due either to first or second order multifractal phase transitions (Lovejoy and Schertzer, 2013). To mitigate these effects (cfr. Supplementary Information) since we deal with the investigation of scale-dependent fractal

dimensions, we evaluate the cumulative statistics at different scales and we observe that extreme fluctuations follow a power law decay leading to the divergence of the 6–th order and the 4–th order moment for $C = 0.008$ and $C = 0.015$, respectively. Thus we fix our range of moments $-6 < q < 6$ and $-4 < q < 4$ for $C = 0.008$ and $C = 0.015$, respectively. This analysis allows characterizing how the (multi)fractal properties of the system evolve with the timescale $\tau$. Indeed, there are ongoing discussions on the fractal structure of both, the atmosphere and the ocean, especially dealing with the short-term variability

and in terms of scaling-law behavior and statistics of increments (e.g., Lovejoy and Schertzer, 2013; Franzke et al., 2020).

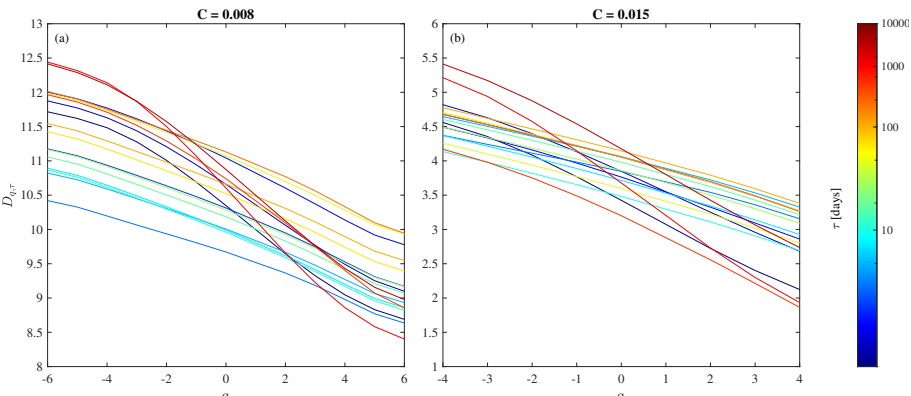

**Figure 9.** $D_{q,\tau}$ spectra for the coupled ocean-atmosphere dynamics at different timescales $\tau_j$ (indicated by different line colors) for reconstructions of MIMFs as in Eq. (12) ($D_{q,\tau}$) for (a) $C = 0.008$ and (b) $C = 0.015$.

The $D_{q,\tau}$ spectrum is reported in Fig. 9, where colored lines correspond to different timescales. It can be observed that for both values of the friction coefficient $C$, different values of $D_{q,\tau}$ are obtained for different $q$, with being $D_{q,\tau}$ a nonlinear decreasing function of $q$. This means that the full system exhibits signatures of multifractality at all timescales, especially at

very short and very long timescales. A simple and direct measure of the degree of multifractality[1] is the so-called multifractal width $\Delta \doteq D_{q_{min},\tau} - D_{q_{max},\tau}$. We observe (see Fig. 10(a,b), black circles) that $\Delta \approx 2$ for $\tau \in [\tau_S, \tau_L]$ days, while $\Delta > 2$ for both $\tau < \tau_S$ and $\tau > \tau_L$, with $\tau_S \sim 20$ days and $\tau_L \sim 1$ year. This behavior could be the reflection of processes operating at different timescales for both the atmosphere (at short timescales) and the ocean (at long timescales). In order to further disentangle those processes, we also evaluated the full spectra of the generalized multifractal dimensions for each subsystem (i.e., atmosphere and ocean) individually. For both values of $C$, the corresponding results are shown in Fig. 11.

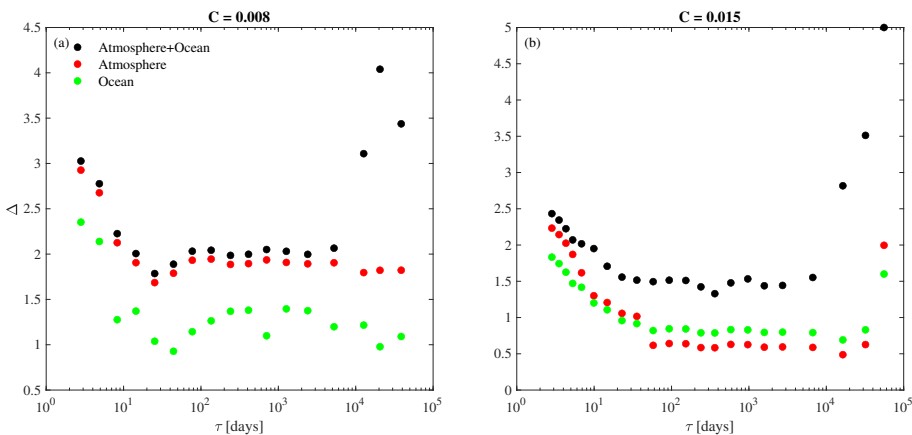

**Figure 10.** Multifractal width $\Delta$ at different timescales $\tau_j$ for reconstructions of MIMFs as in Eq. (12) ($D_{q,\tau}$) for (a) $C = 0.008$ and (b) $C = 0.015$. The different colors refer to the full system (atmosphere+ocean, black circles), only the atmosphere (red circles), and only the ocean (green circles), respectively.

We clearly see that for the atmosphere, there is a scale-independent behavior of $D_{q,\tau}$ for all $q$, rendering the different curves almost invariant with respect to the scale. By contrast, a scale-dependent behavior emerges for the ocean for the lower value of $C$. Indeed, it is evident that as the timescale increases the multiscale generalized dimensions tend to decrease for all values of $q$, moving from $D_{q,\tau_1} \in [5,8]$ to $D_{q,\tau_{17}} \in [2,3]$ for $C = 0.008$. Conversely, although there is an overall reduction in the $D_{q,\tau}$ values for $C = 0.0015$ with respect to those evaluated for $C = 0.008$, the decrease with the timescale is less evident for this higher $C$ value, although it is still present for $\tau > 1$ year (see orange and red curves in comparison with the blue ones in Fig. 11(d)). This clearly suggests that the presence of strong multifractality in the full system can be essentially attributed to the atmosphere, with only a marginal role of the ocean variability in determining the fractal structure of the full system. By evaluating the difference between $D_{q_{min},\tau}$ and $D_{q_{max},\tau}$ we can clearly see that larger values, of the order of 3, are found for the atmosphere, at almost all timescales (and especially at shorter timescales), for both values of $C$. Conversely, larger values are found at shorter timescales for both values of $C$ for the ocean. As the timescale increases, this difference tends to be reduced

---

[1]Another direct measure is the so-called co-dimension of the mean $c = d - D_0$ where $d$ is the dimension of the phase space (e.g., Lovejoy and Schertzer, 2013). For the sake of simplicity we prefer to use here only the multifractal width since it can be easily derived from $D_{q,\tau}$ and not to introduce an additional alternative concept.

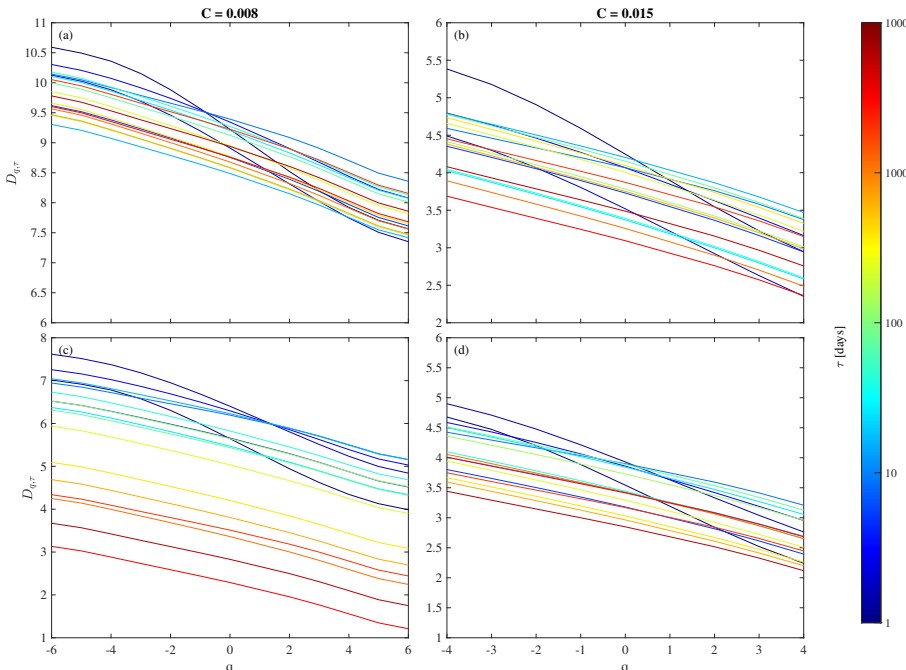

**Figure 11.** $D_{q,\tau}$ spectra for the dynamics of atmosphere and ocean individually at different timescales $\tau_j$ (indicated by different line colors) for reconstructions of MIMFs as in Eq. (12) ($D_{q,\tau}$) for (a,c) $C = 0.008$ and (b,d) $C = 0.015$. Panels (a,b) refer to the atmosphere, (c,d) to the ocean.

to values close to 1, suggesting a reduced multifractality of the ocean with respect to the atmosphere, especially for the lower value of $C$ at larger timescales when the role of the ocean becomes dominant as compared to the atmosphere (see Fig. 2).

### 4.3 Comparison with regional averages from reanalysis data

As a final step we compare our previous results for the reduced order coupled ocean-atmosphere model with those obtained
from reanalysis data (Poli, 2015). More specifically, we use three different sets of regional time series based on the European Centre for Medium-range Weather Forecasts (ECMWF) ORA-20C project (De Boisséson and Balmaseda, 2016; De Boisséson et al., 2017) that is a 10-member ensemble of ocean reanalyses covering the complete 20th century using atmospheric forcing from the ERA-20C reanalysis (https://www.ecmwf.int/en/forecasts/datasets/reanalysis-datasets/era-20c). Here, we focus on data from January 1958 to December 2009 at monthly resolution in terms of different monthly-averaged time series, the set of
data also used previously in Vannitsem and Ekelmans (2018). This period has been chosen in the latter study because of the ocean reanalysis dataset showing here smaller uncertainties than during the first half of the 20th century (De Boisséson and Balmaseda, 2016).

Three different representative regions are chosen: the North Atlantic region, corresponding to the domain defined by $\lambda \in [55°W, 15°W]$ and $\phi \in [25°N, 60°N]$, the North Pacific region, i.e., a spherical-rectangle domain with $\lambda \in [165°E, 225°E]$ and

410 $\phi \in [25°N, 60°N]$, and the Tropical Pacific region, corresponding to $\lambda \in [165°E, 225°E]$ and $\phi \in [25°S, 25°N]$ (Vannitsem and Ekelmans, 2018). The individual series for the two extratropical regions have been derived by projecting the reanalysis fields on two dominant Fourier modes: (i) $F_1 = \sqrt{2}\cos(\pi y/L_y)$, and (ii) $\phi_2 = 2\sin(\pi x/L_x)\sin(2\pi y/L_y)$ (Vannitsem and Ekelmans, 2018). For the Tropical Pacific region, the series are formed by spatial averages. In this way, we obtain two sets of three time series each for both the North Atlantic and the North Pacific (i.e., one for the atmosphere and two for the ocean), and a third

set of three time series for the Tropical Pacific (two for the atmosphere at two different pressure levels and one for the ocean). This allows us to build a 3-D projection of the local atmosphere-ocean coupled dynamics for each region (see Vannitsem and Ekelmans, 2018, for more details).

By using the MEMD analysis to investigate the multivariate patterns of reanalysis data we found the same number of $N_j = 9$ MIMFs for each region, whose mean timescales range from $\sim 2$ months up to $\sim 20$ years, suggesting the existence of multiscale

variability over a wide range of scales. As for the reduced order model, we first investigate the behavior of the spectral energy content $S(\tau)$ of the different MIMFs as a function of their mean timescales $\tau$ as in Eq. (7) for the three different regions as shown in Fig. 12. We clearly observe an increase of the spectral energy content up to a timescale $\tau \sim 1$ year for all regions,

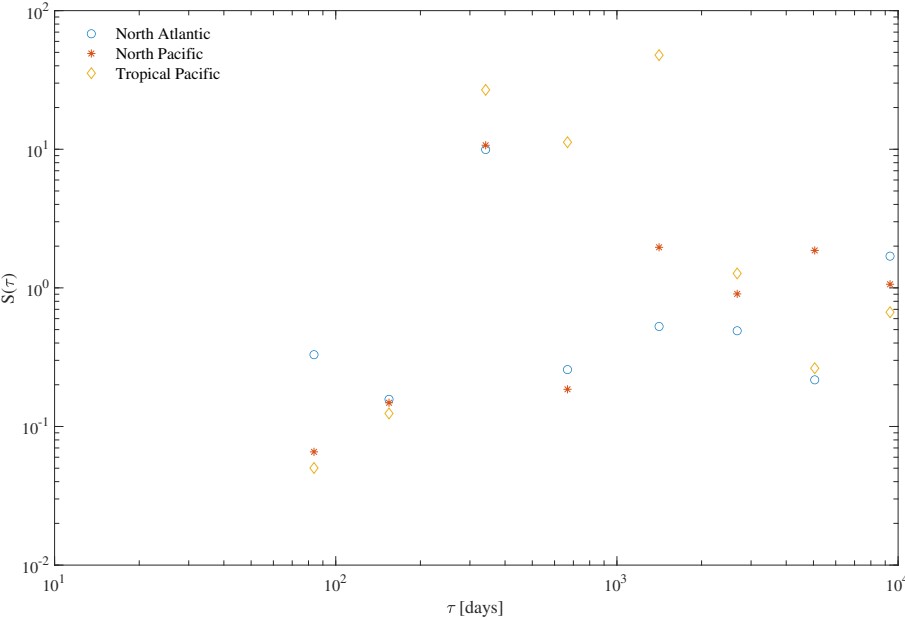

**Figure 12.** Spectral energy content $S(\tau)$ of the different MIMFs as a function of their mean timescales $\tau$ as in Eq. (7) for the North Atlantic (blue circles), the North Pacific (orange asterisks), and the Tropical Pacific (yellow diamonds).

then declining for both the North Atlantic and the North Pacific. Conversely, the Tropical Pacific is characterized by larger spectral content also for timescales larger than 1 year, up to $\tau \sim 5$ years, which coincide with the typical timescales of the El

Niño–Southern Oscillation (ENSO). Furthermore, for all regions a decreasing spectral energy content is found at the largest timescales (i.e., $\tau > 5$ years).

To further compare our above model results with those obtained for the reanalysis data, we evaluate the multiscale generalized fractal dimensions for the three different regions. For each region, we derive both the multifractal width $\Delta \doteq D_{q_{min},\tau} - D_{q_{max},\tau}$ and the full multiscale multifractal spectrum at different timescales $\tau_j$ for reconstructions of MIMFs as in Eq. (12) ($D_{q,\tau}$). Figure 13 shows the corresponding results for the North Atlantic region, the North Pacific region, and the Tropical Pacific region, respectively.

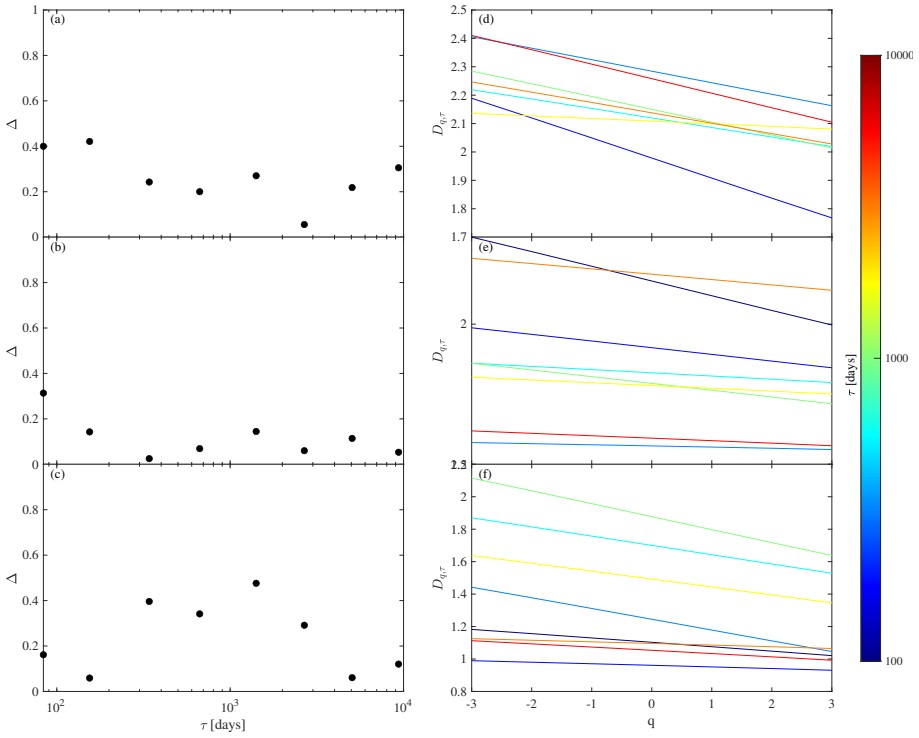

**Figure 13.** (a)-(c) Multifractal width $\Delta$ and (d)-(f) $D_{q,\tau}$ spectra at different timescales $\tau_j$ for reconstructions of MIMFs as in Eq. (12) ($D_{q,\tau}$) for (a,d) the North Atlantic, (b,e) the North Pacific, and (c,f) the Tropical Pacific, respectively.

First of all, it is important to underline that the multiscale generalized fractal dimensions are clearly different with respect to those obtained from the ocean-atmosphere model. This directly follows from the different numbers of variables (time series) in the model, being a 36-dimensional dynamical system, with respect to the reanalysis data, being a 3-dimensional projection of the regional ocean-atmosphere dynamics. Nevertheless, although different in terms of absolute values, both the model and the reanalysis data show a similar qualitative behavior with varying scale $\tau$, although some differences are found between the different regions.

On one hand, both the North Atlantic and the North Pacific regions (see Fig. 13(d,e)) are characterized by a scale-dependent behavior, with decreasing $D_{q,\tau}$ as $\tau$ increases. Moreover, by looking at the multifractal width as a function of the scale (Fig. 13(a,b)) we find evidence for a decreasing $\Delta$ as $\tau$ increases, being representative of a transition from a short-term multi-

fractal nature to a long-term monofractal one. These features can be interpreted in terms of the different multiscale dynamical processes affecting the atmosphere on short scales and the ocean on larger scales.

On the other hand, by looking at the Tropical Pacific region we clearly see an enhancement of $\Delta$, i.e., the emergence of multifractal features (see Fig. 13(c)), at annual/multi-annual timescales (i.e., $\tau \sim 1 - 8$ years), being also characterized by the largest values of the multiscale generalized fractal dimensions (see Fig. 13(f)). This could be related to the role of the El Niño–Southern Oscillation (ENSO) cycle manifesting at these timescales (between 2 and 7 years), which is likely responsible for the different scale-dependent behavior of $D_{q,\tau}$ as compared to the two other extratropical regions.

In summary, by means of the reanalysis data, we have been able to demonstrate that i) the reduced order coupled ocean-atmosphere model and the reanalysis data show some qualitatively similar behavior of the multiscale generalized fractal dimensions, although characterized by different absolute values due to the different numbers of variables considered in the model and the projections on a few modes of the reanalysis data, and that ii) interesting features emerge when looking at the scale-dependency of the statistics of the phase-space scaling for different regions, being the reflection of different driving mechanisms and processes operating at different timescales in the coupled ocean-atmosphere system. However, further investigations are needed to characterize the role of the different processes as well as their intrinsic dimensionality, occurrence, and spatial dependency in more detail. Such an in-depth investigation is outlined as a part of our future work.

## 5 Conclusions

We have provided a first time systematic investigation of the multiscale dynamics of a reduced order coupled ocean-atmosphere model (Vannitsem et al., 2015) as described by means of the statistics of the phase-space scaling (Alberti et al., 2020a).

First, by means of the Multivariate Empirical Mode Decomposition (MEMD) we have been able to detect oscillating patterns with time-dependent amplitude and phase that are directly linked to a rich variety of features of the coupled ocean-atmosphere system. We have found that the underlying structure of the 3-D projection of the full attractor is essentially reproduced by a subset of Multivariate Intrinsic Mode Functions (MIMFs) corresponding to the most relevant timescales without too much loss of information, thus further reducing the complexity of the reduced order model itself. These results appear relevant if put into the wider context of coupled ocean-atmosphere dynamics, allowing us to recover the main features by only considering the most relevant (in terms of energy) timescale dynamical components.

Second, by exploiting the novel concept of multiscale/multivariate generalized fractal dimensions we have investigated the different multifractal properties for the ocean and the atmosphere at different timescales. We have demonstrated that for weak ocean-atmosphere coupling (i.e., for low values of the friction coefficient $C$), the resulting dimensions of the two model components are very different, while for strong coupling (larger $C$) at which coupled modes develop at low frequencies, the scaling properties are more similar especially at longer time scales. These results suggest that as $C$ increases, we observe the development of a coherent coupled dynamics, primarily at large timescales. In terms of the underlying fractal structure, we have found that for both considered values of the friction coefficient $C$, the full system exhibits signatures of multifractality at all timescales, especially pronounced at short and long as compared to intermediate timescales. By means of the full spectrum

of generalized fractal dimensions, we have clearly evidenced that for the atmosphere, there is a scale-independent behavior of $D_{q,\tau}$ for all $q$, rendering the multifractal spectra almost invariant with respect to the timescale. By contrast, a scale-dependent behavior emerges for the ocean for the lower value of $C$. This clearly suggests that the presence of strong multifractality in the full system can be attributed to the atmosphere, with only a marginal role of the ocean variability in determining the fractal structure of the full system.

Finally, we have compared our results for the reduced order coupled ocean-atmosphere model with those derived from reanalysis data (Poli, 2015) by using three sets of different regional time series from the ORA-20C project (De Boisséson and Balmaseda, 2016; De Boisséson et al., 2017). Although the resulting multiscale generalized fractal dimensions clearly differ quantitatively from those obtained from the ocean-atmosphere model – which can be easily understood by considering the different dimensions of the model (a 36-dimensional dynamical system) and the reanalysis data (3-dimensional projections of the local ocean-atmosphere dynamics) – we observed a similar qualitative behavior with changing scale $\tau$. Interestingly, the multiscale multifractal features of different regions show different scale-dependent behaviors. Specifically, both the North Atlantic and the North Pacific regions are characterized by a scale-dependent behavior, with decreasing $D_{q,\tau}$ as $\tau$ increases, with a transition from a short-term multifractal nature to long-term monofractal one. These features can be interpreted in terms of the different multiscale dynamical processes affecting the atmosphere at short timescales and the ocean at longer timescales. Conversely, the Tropical Pacific region is characterized by the emergence of multifractal features at annual/multi-annual timescales (i.e., $\tau \sim 1 - 8$ years), being also characterized by the largest values of the multiscale generalized fractal dimensions. This behavior can be seen as a manifestation of the El Niño–Southern Oscillation (ENSO) cycle that typically acts at these timescales and can be considered the key driving factor of a different scale-dependent behavior of $D_{q,\tau}$ as compared to the two extratropical regions.

Our findings for both the model and the reanalysis data suggest that our approach can be used to diagnose the strength of coupling in the ocean-atmosphere system and to investigate the statistics of the phase-space scaling of the system. We have demonstrated that the model and the reanalysis data show a qualitatively similar behavior of the multiscale generalized fractal dimensions. However, the different scale-dependency of the statistics of the phase-space scaling for different regions can contribute to a better understanding of the different driving mechanisms and processes operating at different timescales in the coupled ocean-atmosphere system. Indeed, our results highlight that the complexity of the coupled ocean-atmosphere system significantly depends not only on model parameters, that can be helpful for reproducing different features of the dynamics, but also on the particular scale we are looking at that can be related to different phenomena and source mechanisms, of both intrinsic and external origin to the ocean-atmosphere system. This means that our results could be also helpful for understanding the dimensionality of the system at different time scales, thus being useful for forecasting the dynamics at different scales and for building empirical models based on dynamical system approaches in a similar fashion to models developed considering real space scaling behavior (e.g., Del Rio Amador and Lovejoy, 2021a, b). These observations suggest that further investigations are needed to better characterize the role of the different processes as well as their intrinsic dimensionality, occurrence, and spatial dependency, which shall be further addressed in our future work.

*Code availability.* All codes used for the analysis and generating the figures can be obtained from the authors upon request.

*Data availability.* The time series of the model used in the present manuscript are available from the authors upon request. The reanalysis
dataset is available on Zenodo, https://doi.org/10.5281/zenodo.1135134.

*Author contributions.* TA, RVD, and SV designed the study. TA conducted the analysis and drafted the manuscript. RVD and SV contributed to the interpretation of the results. All authors contributed to the writing of the manuscript.

*Competing interests.* The authors declare that they have no conflict of interest.

*Acknowledgements.* This work has been partially supported by the German Federal Ministry for Education and Research (BMBF, grant no.
01LP2002B) and the Belgian Science Policy Office (Belspo) through the JPI Climate/JPI Oceans project ROADMAP.

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
