# Peer review of "Multiscale fractal dimension analysis of a reduced order model of coupled ocean-atmosphere dynamics"

_Earth System Dynamics, 2020_

## Referee Comment (RC2)

**Comments on "Multiscale fractal dimension analysis of a reduced order model of coupled ocean-atmosphere dynamics"**

Tommaso Alberti, Reik V. Donner, and Stéphane Vannitsem

General comments:

The authors propose combining two apparently contradictory analysis techniques to the outputs of a low (36) dimensional dynamical ocean - atmosphere model. The first, makes a nontrivial decomposition of the 36 dimensional signal into series with well-defined time scales, the second analyses the phase spaces assuming the existence of scale invariant properties. The justification and interpretation of this is opaque.

While the authors question the utility of conventional analysis techniques, at least the latter can be interpreted in straightforward manners. The interpretation of their results is nontrivial.

Detailed comments:

The notation is not easy to follow. Please explain the curly bracket notation used throughout:

$$\{\mathbf{s}(t)\}|_{t \in T} = \{s_1(t), s_2(t), \ldots, s_k(t)\}$$

On the left, a bold symbol "**s**" is used which is standard for indicating a vector. Why do the authors (apparently needlessly) add curly brackets and then an explicit restriction as a subscript?

Further, there is the bizarre looking symbol $(D_2^{\Sigma_j})$, that is also not adequately explained.

When discussing the mathematical properties of the usual decompositions ("completeness, convergence, linearity, and stationarity") it is stated that "these conditions are not usually met when real-world geophysical data are analyzed". This is confusing since the mathematical properties of Fourier or other decompositions are valid irrespective of any application. I think the authors meant to question the appropriateness of such decompositions for their specific application? However, this is a mathematical question that cannot be answered without reference to a specific assumed mathematical framework. In the paper the authors do not analyze empirical data at all but rather model outputs. Contrary to real empirical series, their series are therefore taken from a well-defined mathematical framework given by dynamical systems theory. Please explain why standard decompositions are not adequate for studying such model outputs and why there is a need for them to be replaced by decompositions with quite nontrivial interpretations and properties.

Also in the Methods section, it is stated that the authors "put forward a novel approach based on combining two different data analysis methods for Multivariate Empirical Mode Decomposition and generalized fractal dimensions". What is confusing is that while the MEMD analyzes time series in real space, in their application, the generalized fractal dimensions analysis is carried out in a quite different space - the phase space of each series. The result is that for each time series with characteristic time scale τ, that the corresponding phase spaces are assumed to be scaling. In other words, while there are essentially no scaling properties in real space, it is assumed that there will be nontrivial scaling properties in the corresponding phase space. The approach is presumably justified if the characterizing these scaling properties via generalized fractal dimensions will help understand the system. At this point one wonders whether the conventional Fourier spectrum of each τ scale series might have been easier to interpret, to understand. All this needs explanation, clarification.

In particular, when the generalized fractal dimensions are estimated, the authors need to show that there are indeed some phase space scaling properties. Using mathematical definitions such as eqs. 1-3 - where the small scale limits are taken - has only a formal validity when the definitions are applied to numerical model outputs, especially when the latter has been subjected to cubic spline interpolation which makes the small scales artificially smooth. In practice, one needs to display scaling behaviour over at least an order of magnitude or so in scale in order for any fractal dimension estimates to be convincing. The authors must therefore display some of their scaling plots - not just logarithmic slopes that have already been interpreted in terms of dimensions.

In this regard, I could also add that figs. 9 and 11 are almost certainly largely spurious. This is because typically for moments of order $q\approx>3\text{-}4$, the moments are completely dominated by a single hypercube (a "second order multifractal phase transition") so that for larger $q$, the values will depend sensitively on the exact details of the input series. Similarly for $q<0$ most if not all the values will likely be spurious essentially due to the statistics of the very sparsely populated regions of phase space (the very low probability regions, see e.g. the discussion in ch. 5 of [$Lovejoy\ and\ Schertzer$, 2013]). In other words over most of the range of moments given in the figure ($-20<q<20$), the dimensions are likely to be spurious .

Finally, the interpretation of the key figures 5-8 is not at all obvious. Calling these characterizations "topological, geometric" is unhelpful and/or misleading since they are actually statistical exponents without any straightforward relationship to the phenomenon under study.

The authors could note that whereas a white noise signal would give a correlation dimension equal to the dimension of the phase space itself (it is space filling), that a Brownian motion in a space $d\geq2$ has a constant dimension = 2.

**Reference:**

Lovejoy, S., and Schertzer, D., *The Weather and  Climate: Emergent Laws and Multifractal Cascades*, 496 pp., Cambridge University Press, 2013.

---

## Author Comment (AC1)

*We really appreciate the positive evaluation of our manuscript and the nice words the Referee used for describing both our approach and our findings.*
*We thank the Referee for raising some points that can be helpful for improving the presentation and clarity of our findings.*
*Here we provide short replies (in italics, labelled by "A") to the Referee's comments (in normal font, labelled by "C") that will be addressed in a more detailed way in our final response and thoroughly considered in a revised version of our manuscript.*

**C1.** The work by Alberti et al. is a very intensive and information paper. It shows how to model Atmosphere and ocean dynamics within the scope of ESD. The authors extended the concept of multiscale generalized fractal dimensions employing Multivariate Empirical Mode Decomposition to analyze multiscale and multivariate behavior of the ocean-atmosphere coupled dynamics. Although the concept is not new to the scientific community, it is interesting to know how such a process is applicable for elucidating atmospheric behavior. The one important thing is that they tried to give more credits to the relevant works as much as possible.
The paper is well written with proper usage of English and scientific jargon. However, for the general audience, some of the terminologies need to be explained simpler. For example, the readers may not necessarily need to know about the Hausdorff dimension.

**A1.** *We will be more precise when introducing some concepts and terms (as for example the notion of Hausdorff dimension or generally the fractal dimensions themselves).*

**C2.** Although they are making some valid assumptions in the methodology, some statements are a bit confusing. For instance, the authors mentioned that mathematical properties of completeness, convergence, linearity, and stationarity are usually not met when real-world geophysical data are analyzed. But it is not clear the reason behind this and what makes the use of adaptive methods. How is the complexity of data suitable for such methods?. Likewise, while the shifting process needs careful implementation for multivariate techniques, Mandic (2010) proposed an alternative way to cubic spline interpolation in each direction with a quasi-Monte Carlo-based approach. But the reviewer does not fully agree with it as such interpolation may lose the data's intrinsic properties since this approach produces smoother dynamics that do not exist in the data.

**A2.** *We really appreciate this comment that allows us to make more clear some statements, trying to reduce some possible confusion.*
*We are ready to clarify the sentence on properties met by real-world data since linearity and stationarity are usually not met. This is also strictly related to the use of adaptive methods that can be justified to overcome some limitations of fixed-basis methods such as linearity and stationarity assumption. Moreover, adaptive methods (as the MEMD) could be more suitable for reducing some mathematical assumptions and a priori constraints.*
*Concerning the quasi-Monte Carlo-based approach, it is used only to provide a more uniform set of direction vectors over which to compute the local mean of envelopes, and not to interpolate maxima and minima and/or to manipulate the data introducing a smoother dynamics. Moreover, the quasi-Monte Carlo method is also needed to avoid implicitly preferred directions that could be more dominant with respect to the others, which could introduce a source of errors in evaluating signal projections. We will add more details and corrections in a revised version of the manuscript to be more clear and to avoid confusion.*

**C3.** The authors tried to interpret most of the results efficiently. However, some of the interpretation is very unclear and hard to follow. For example, the authors did not mention what is the physical meaning behind the correlation dimension. It is just a kind of statistics of the data. Without understanding the physical meaning, it is not clear why it is a function of time. Another issue is that some of the figures are not interpreted well. e.g., the description of Figures 5 and 6 are not marched. They are not clear, as seen in the figures. For the general audience, they are confusing. Even though the multiscale correlation dimension for each MIMF decreases with an increasing timescale, as seen in panel (a), the other two panels are not well elaborated.

**A3.** *We thank the Referee for this comment. $D_0$, $D_1$, and $D_2$ are strictly related to different properties of physical systems: (i) one purely geometric measure ($D_0$) providing us information on the coverage of the phase-space by the studied system's dynamics, (ii) one information measure ($D_1$) giving us a measure of the information gained on the phase-space with a given accuracy $\varepsilon$, and (iii) one measure of correlations, i.e., mutual dependence, between phase-space points ($D_2$). Since the collective behavior of a system is given by physical processes operating at different scales, it is straightforward to look how they contribute to the topology of the phase-space, not only singularly (as in panel (a) of Figs. 5-6) but especially when considering all processes occurring below a selected scale (as in panel (b) of Figs. 5-6) and by looking separately at the atmosphere and ocean (as in panel (c) of Figs. 5-6).*
*In a revised version of our manuscript, we will provide more details on the fractal dimensions in terms of their physical meaning (at least for $D_0$, $D_1$, and $D_2$), together with a more detailed description of Figures 5-6.*

**C4.** The authors cleverly described the experiments. To reproduce the work, one needs to understand all the mathematical formulas. In the scientific method, some time calculation and mathematical expression do not match as most of the calculation procedures follow fundamental statistical programming. It needs a concise explanation of calculating all these quantities like system attractors, phase space, and correlation dimensions. The description of these quantities introduced in the manuscript is very dubious and complex to replicate. The reviewer is thankful for providing data sets. But it becomes worthy if it includes an explanation of how to reanalyze these data sets.

**A4.** *We thank the Referee for this suggestion. Our full system consists of 36 variables, thus we are working on a 36-D space. For visual purposes, we reduced our 36-D space to a 3-D subspace by looking at the behavior of the three selected dynamical variables (i.e., $T_{o,2}$, $\Psi_{o,2}$, and $\psi_{a,1}$) in a 3D plot. This allows us to investigate the 3D projection of the full system phase-space attractor, i.e., the set of values toward which our system tends to evolve.*
*Concerning the calculation of the generalized fractal dimensions, we used the approach proposed by Hentschel and Procaccia (1983) consisting of partitioning the phase-space into hypercubes and then measuring the probability of finding a given hypercube filled by points and/or its generalization to a statistical order q (as also described at lines 40-52 of the submitted version). We will also be more precise in a revised version and include more details on the calculations of $D_q$.*

**C5.** Finally, the reviewer appreciates the work of the authors. Still, it needs a bit more simplification and incorporating the issues mentioned above.

**A5.** *We really thank the Referee for his/her nice words on our manuscript and we will do our best to be more precise and more clear in the revised version of the manuscript.*

---

## Author Comment (AC2)

General comments:

The authors propose combining two apparently contradictory analysis techniques to the outputs of a low (36) dimensional dynamical ocean - atmosphere model. The first, makes a nontrivial decomposition of the 36 dimensional signal into series with well-defined time scales, the second analyses the phase spaces assuming the existence of scale invariant properties. The justification and interpretation of this is opaque.
While the authors question the utility of conventional analysis techniques, at least the latter can be interpreted in straightforward manners. The interpretation of their results is nontrivial.

*We thank the Referee for raising some points that can be helpful for improving the presentation and clarity of our findings. Most of all, we do not question the utility of conventional analysis techniques, but acknowledge their intrinsic limitations and attempt to explore the potentials of a combination of two "non-conventional" techniques to provide additional information.*
*We also want to stress that the two methods should not be seen as "apparently contradictory" as emphasized by the reviewer. The modes extracted in the first analysis step have no well-defined time scales but are instead characterized by scales that are time-dependent. This is one of the main novelties of the Empirical Mode Decomposition and its multivariate extension we used here (i.e., the MEMD) as compared to fixed-scale decomposition methods like wavelets. The extracted modes can be seen as representative of fluctuations at a typical scale that is the average of the instantaneous scales derived from a given mode via the Hilbert Transform. Moreover, the second analysis step, i.e., the generalized fractal dimensions, requires to have scale invariant properties in the phase-space of a given system, thus working (essentially) on measuring the geometrical properties of the system trajectory and information on how to reconstruct it by measuring the information dimension $D_1$ and q-tuplet correlations $D_{q>1}$. This means that there are no a priori constraints on understanding a system using $D_q$. Thus, the two methods are not contradictory but rather complementary.*

*In the following we provide replies (in italics, labelled by "A") to the Referee's detailed comments (in normal font, labelled by "C") that will be also thoroughly considered in a revised version of our manuscript.*

Detailed comments:
**C1.** The notation is not easy to follow. Please explain the curly bracket notation used throughout:

$$\{s(t)\}|_{t \in T} = \{s_1(t), s_2(t), \ldots, s_k(t)\}$$

On the left, a bold symbol "s" is used which is standard for indicating a vector. Why do the authors (apparently needlessly) add curly brackets and then an explicit restriction as a subscript?

Further, there is the bizarre looking symbol $(D_2^{\Sigma_j})$, that is also not adequately explained.

**A1.** *We thank the Referee for this suggestion. Indeed, we agree that the notation using curly brackets has been partly misleading, since the left-hand side of the equation was originally intended to represent a sequence of vectors, while the right-hand side was supposed to clarify the structure of each of those vectors composed of k scalar properties, the latter of which however was lacking clarity in our notation. This aspect will be clarified in our revised manuscript. Moreover, the "bizarre looking" symbol can be safely changed to $D_{q,\tau}$ following the notation used in Alberti et al.*

*(Chaos, 2020). We will modify the corresponding parts of our manuscript also with a general attempt to be more precise when introducing notations in a revised version of our manuscript.*

**C2.** When discussing the mathematical properties of the usual decompositions ("completeness, convergence, linearity, and stationarity") it is stated that "these conditions are not usually met when real-world geophysical data are analyzed". This is confusing since the mathematical properties of Fourier or other decompositions are valid irrespective of any application. I think the authors meant to question the appropriateness of such decompositions for their specific application? However, this is a mathematical question that cannot be answered without reference to a specific assumed mathematical framework. In the paper the authors do not analyze empirical data at all but rather model outputs. Contrary to real empirical series, their series are therefore taken from a well-defined mathematical framework given by dynamical systems theory. Please explain why standard decompositions are not adequate for studying such model outputs and why there is a need for them to be replaced by decompositions with quite nontrivial interpretations and properties.

**A2.** *We thank the Referee for this important suggestion. As also highlighted in our reply to Referee #1 we need to clarify the sentence on properties met by real-world data (note that our manuscript does not exclusively utilize low-order model output, but also reanalysis data, which in our opinion would qualify as "empirical") for which linearity and stationarity assumptions are often not met. Indeed, we fully agree that mathematical properties of the decomposition methods themselves are surely valid irrespective of any application. As suggested by the Referee, we referred to the use of adaptive methods that can be justified to overcome some limitations of fixed-basis methods such as linearity and stationarity assumptions. Moreover, adaptive methods (as the MEMD) could be more suitable for reducing some mathematical assumptions and a priori constraints.*
*Although we use the MEMD on a well-defined framework derived from dynamical systems theory, the reduced a priori constraints and the limited number of intrinsic components that can be visually inspected could be an advantage with respect to standard decompositions. Another advantage concerns the combination with generalized fractal dimensions: if we, for example, use Fourier decomposition we will have a large number of (harmonic) oscillating components at different fixed frequencies that should be summed up for exploiting our proposed procedure. Furthermore, if we, for example, use wavelets we will deal with some a priori assumptions on the decomposition basis onto which we are projecting our data that could produce misleading results in our procedure of evaluating fractal measures on a priori fixed scales. Thus, we do not question the appropriateness of conventional analysis techniques, but rather acknowledge that they (as well as any other) have intrinsic limitations in what we can learn from them.*

**C3.** Also in the Methods section, it is stated that the authors "put forward a novel approach based on combining two different data analysis methods for Multivariate Empirical Mode Decomposition and generalized fractal dimensions". What is confusing is that while the MEMD analyzes time series in real space, in their application, the generalized fractal dimensions analysis is carried out in a quite different space - the phase space of each series. The result is that for each time series with characteristic time scale t, that the corresponding phase spaces are assumed to be scaling. In other words, while there are essentially no scaling properties in real space, it is assumed that there will be nontrivial scaling properties in the corresponding phase space. The approach is presumably justified if the characterizing these scaling properties via generalized fractal dimensions will help understand the system. At this point one wonders whether the conventional Fourier spectrum of each t scale series might have been easier to interpret, to understand. All this needs explanation, clarification.

**A3.** *We really appreciate this comment since it allows us to better underline our main aim. We are interested in investigating how phase-space properties (geometry, correlations) change when dynamical components at different mean scales with different dynamics are considered. In other words, we are interested in looking at the role of scale-dependent phenomena in defining the whole properties of a system. Global measures proposed in the past only allow us to investigate the statistical, topological, geometrical, scaling, properties of the whole system; conversely, our proposed approach allows us to investigate how the different scales contribute to the global properties of a system. Moreover, our framework also provides consistency with established measures for characterizing time series from an integral (not scale-resolved) perspective, since the scale-dependent measures we evaluate converge to the associated global measures as all scales are considered, i.e., when the full system dynamics, composed by all accessible scales, is reached, Within this framework, our approach could be promising for investigating scale-dependent properties, as measured by fractal dimensions, of the system. We are indeed interested in nonlinear variability characteristics at different time scales, thus employing for example Fourier decomposition would leave us with perfectly linear and stationary harmonic functions as components, which do not carry any information on nonlinear dynamics, unless when studying their mutual phase relationships, leaving out the high-order statistical properties and only focusing on the autocorrelation function (i.e., the second-order moment). Otherwise, by looking at the behavior of fractal dimensions we can explore how the different scales contribute to change the phase-space properties that cannot be highlighted by using the conventional Fourier spectrum.*

**C4.** In particular, when the generalized fractal dimensions are estimated, the authors need to show that there are indeed some phase space scaling properties. Using mathematical definitions such as eqs. 1-3 - where the small scale limits are taken - has only a formal validity when the definitions are applied to numerical model outputs, especially when the latter has been subjected to cubic spline interpolation which makes the small scales artificially smooth. In practice, one needs to display scaling behaviour over at least an order of magnitude or so in scale in order for any fractal dimension estimates to be convincing. The authors must therefore display some of their scaling plots - not just logarithmic slopes that have already been interpreted in terms of dimensions.

**A4.** *We agree with this comment. To be clearer and more convincing, we display in Figs. 1 and 2 of this response letter (which will also be included in a revised version of our manuscript) the scaling behavior for the correlation integral for the two cases C=0.008 and C=0.015 at different timescales. We choose to show here only the correlation integral since it can be faster evaluated than other moments (cfr. Grassberger and Procaccia, 1983). We show here that there exists at least an order of magnitude in scale over which a scaling behavior is observed. A similar behavior is also observed when considering the reanalysis data as shown in Fig. 3 of this response letter for the different regions. Taking also into consideration a comment by Referee #1, we consider adding Supplementary Materials with more details on the computation of fractal dimensions and scaling plots to a revised version of our manuscript.*

*We would further like to remark that the cubic spline interpolation does not produce artificially smoothed small scales since it does not act on the data themselves but only on local extreme values of the data to extract intrinsic oscillating components from the data. Thus, the shape of the raw data is not changed and generally the (M)EMD extracts scale-dependent components that are smoother as the largest scales are approached.*

[Figure]

**Fig. 1** The log-log scaling plots of the correlation integral C(r) as a function of r (normalized with the respect to the largest possible separation between points in the phase-space represented by $r_0$) at different scales represented by colors for the case C=0.008. The lines refer to the power law fit in the limit r→0.

[Figure]

**Fig. 2** The log-log scaling plots of the correlation integral C(r) as a function r (normalized with the respect to the largest possible separation between points in the phase-space represented by $r_0$) at different scales represented by colors for the case C=0.015. The lines refer to the power law fit in the limit r→0.

[Figure]

**Fig. 3** The log-log scaling plots of the correlation integral C(r) as a function r (normalized with the respect to the largest possible separation between points in the phase-space represented by $r_0$) at different scales represented by colors for the reanalysis data.
The lines refer to the power law fit in the limit r→0.

**C5.** In this regard, I could also add that figs. 9 and 11 are almost certainly largely spurious. This is because typically for moments of order q≈>3-4, the moments are completely dominated by a single hypercube (a "second order multifractal phase transition") so that for larger q, the values will depend sensitively on the exact details of the input series. Similarly for q<0 most if not all the values will likely be spurious essentially due to the statistics of the very sparsely populated regions of phase space (the very low probability regions, see e.g. the discussion in ch. 5 of [Lovejoy and Schertzer, 2013]). In other words over most of the range of moments given in the figure (-20<q<20), the dimensions are likely to be spurious.

**A5.** *We thank the Referee for raising this important point on the statistical significance of higher-order moments. We are aware that this is a crucial point, especially when working with scale invariant features measured via structure functions, detrended fluctuation analysis, and spectral methods (as for wavelets). To deal with this problem and to support the statistical significance of our results we have followed the approach also described in Ch. 5 of Lovejoy and Schertzer (2013) to evaluate the maximum moments as those derived from the tail of the cumulative distribution function of the data. Since we deal with the investigation of scale-dependent fractal dimensions, we evaluate the cumulative statistics at different scales and as shown in Figs. 4-6 in this response letter we observe that extreme fluctuations follow a power law decay leading to the divergence of the 6th-order moment and the 4th-order moment for C=0.008 and C=0.015, respectively. Thus we fix our range of moments -6<q<6 and -4<q<4 for C=0.008 and C=0.015, respectively, and we will modify accordingly Figs. 9-11 in a revised version of our manuscript. Similar results are also obtained for the reanalysis data (see Figs. 7-8 in this response letter), thus we fix here our range of moments to -3<q<3.*

[Figure]

**Fig. 4** The cumulative distribution function at different scales as reported by different colors for the case C=0.008. The lines refer to the power law fit of the tail.

[Figure]

**Fig. 5** The cumulative distribution function at different scales as reported by different colors for the case C=0.015. The lines refer to the power law fit of the tail.

[Figure]

**Fig. 6** The power-law scaling exponent $q_D$ as a function of the different scales for the case C=0.008 (black asterisks) and C=0.015 (red diamonds). The minimum $q_D$ has been chosen to set the range of statistically significant moments.

[Figure]

**Fig. 7** The cumulative distribution function at different scales as reported by different colors for the reanalysis data. The lines refer to the power law fit of the tail.

[Figure]

**Fig. 8** The power-law scaling exponent q_D as a function of the different scales for the reanalysis data. The minimum q_D has been chosen to set the range of statistically significant moments.

**C6.** Finally, the interpretation of the key figures 5-8 is not at all obvious. Calling these characterizations "topological, geometric" is unhelpful and/or misleading since they are actually statistical exponents without any straightforward relationship to the phenomenon under study. The authors could note that whereas a white noise signal would give a correlation dimension equal to the dimension of the phase space itself (it is space filling), that a Brownian motion in a space $d \geq 2$ has a constant dimension = 2.

**A6.** *We thank the Referee for this comment. We will work on further improving the clarity of our manuscript, especially when introducing some key concepts and/or describing key features. We are referring to topological and geometrical since some measures are able to give us information on phase-space properties. For example, $D_0$ is a measure of the filling of the phase-space, thus providing a measure on the coverage of the phase-space by the studied system's dynamics, $D_1$ provides a measure of the information gained on the phase-space with a given accuracy $\varepsilon$, and the $D_{q>1}$ provide measures of q-tuplet correlations, i.e., mutual dependence, between phase-space points. This explains why we used the terms topological and geometrical in our manuscript.*

Reference:

Lovejoy, S., and Schertzer, D., The Weather and Climate: Emergent Laws and Multifractal Cascades, 496 pp., Cambridge University Press, 2013.

*Thanks a lot for this reference that we will consider in a revised version of the manuscript.*

---

## Author Response (AR1)

Dear Editor,

first of all, we would like to express our sincere thanks for your overall positive evaluation of our manuscript and the highly valuable comments of the two reviewers who have evaluated our work. With this letter, we are submitting a revised version of our manuscript entitled "Multiscale fractal dimension analysis of a reduced order model of coupled ocean-atmosphere dynamics". We carefully considered and addressed all Referees' comments and suggestions to improve our manuscript.

We are confident that the revised version allows us to present our results in a more detailed and appropriate way, improving the clarity and the readability of our manuscript. In the following, we provide a point-by-point reply (in italics) to all comments (in normal font) of both Referees.

Sincerely,
Tommaso Alberti, Reik Donner, and Stéphane Vannitsem

**Referee #1**

**C1.** The work by Alberti et al. is a very intensive and information paper. It shows how to model Atmosphere and ocean dynamics within the scope of ESD. The authors extended the concept of multiscale generalized fractal dimensions employing Multivariate Empirical Mode Decomposition to analyze multiscale and multivariate behavior of the ocean-atmosphere coupled dynamics. Although the concept is not new to the scientific community, it is interesting to know how such a process is applicable for elucidating atmospheric behavior. The one important thing is that they tried to give more credits to the relevant works as much as possible.

The paper is well written with proper usage of English and scientific jargon. However, for the general audience, some of the terminologies need to be explained simpler. For example, the readers may not necessarily need to know about the Hausdorff dimension.

**A1.** *We really appreciate the positive evaluation of our manuscript both in terms of scientific/ methodological and formal points of view. To address some of the Referees' comments we decided to add Supplementary Material in which we introduce more precisely some relevant concepts and terms around the notion of generalized fractal dimensions. This is done at lines 50-60 where the supplementary material is also referenced.*

**C2.** Although they are making some valid assumptions in the methodology, some statements are a bit confusing. For instance, the authors mentioned that mathematical properties of completeness, convergence, linearity, and stationarity are usually not met when real-world geophysical data are analyzed. But it is not clear the reason behind this and what makes the use of adaptive methods. How is the complexity of data suitable for such methods?. Likewise, while the shifting process needs careful implementation for multivariate techniques, Mandic (2010) proposed an alternative way to cubic spline interpolation in each direction with a quasi-Monte Carlo-based approach. But the reviewer does not fully agree with it as such interpolation may lose the data's intrinsic properties since this approach produces smoother dynamics that do not exist in the data.

**A2.** *We really appreciate this comment that allows us to make more clear some statements, trying to reduce some possible confusion.*

*First of all, we thank the Reviewer for highlighting an imprecise sentence that we wrote in our manuscript. Indeed, the mentioned properties that are usually not met in real-world geophysical data should only include linearity and stationarity, and of course not convergence and completeness issues. For this reason, we have corrected this sentence accordingly.*

*Furthermore, adaptive methods can be helpful for overcoming some limitations of fixed-basis methods that are generally characterized by linearity and stationarity assumptions (as for Fourier analysis, for example). In addition, fixed-basis methods implicitly assume that a given natural phenomenon or a superposition of physical processes can be represented in terms of a priori defined mathematical functions like sine and/or cosine or some other kinds of wave functions. This cannot be assured, thus adaptive methods (as the MEMD) could be more suitable for reducing some mathematical assumptions and a priori constraints.*

*Finally, concerning the quasi-Monte Carlo-based approach it is used only to provide a more uniform set of direction vectors over which to compute the local mean of envelopes, and not to interpolate maxima and minima and/or to manipulate the data introducing a smoother dynamics. Indeed the core of the MEMD algorithm proposed by Rehman and Mandic (2010) consists on the following procedures:*

1. *given a n-dimensional space we need to find the direction vectors by considering that these vectors reduce to points in a (n-1)-dimensional space;*
2. *the simplest choice is to employ uniform angular sampling on an n-dimensional hypersphere but this will lead to a non-uniform filling of the n-dimensional space (there would be a higher density of points near the poles of the n-dimensional hypersphere;*
3. *a quasi-Monte Carlo method is then used to provide a more uniform distribution of direction vectors;*
4. *once the direction vectors are chosen, the signal is projected onto these vectors, the extrema of the resulting projected signals are evaluated and interpolated component-wise to yield multidimensional envelopes that are then averaged to obtain the multivariate mean.*

*The quasi-Monte Carlo method is needed only for selecting a uniform sampling of direction vectors, thus to avoid implicitly preferred directions that could be more dominant with respect to the others, which could introduce a source of errors in evaluating signal projections.*

*We have modified both Section 3 "Methods" and Section 3.1 "Multivariate Empirical Mode Decomposition (MEMD)" to add all these details and corrections.*

**C3.** The authors tried to interpret most of the results efficiently. However, some of the interpretation is very unclear and hard to follow. For example, the authors did not mention what is the physical meaning behind the correlation dimension. It is just a kind of statistics of the data. Without understanding the physical meaning, it is not clear why it is a function of time. Another issue is that some of the figures are not interpreted well. e.g., the description of Figures 5 and 6 are not marched. They are not clear, as seen in the figures. For the general audience, they are confusing. Even though the multiscale correlation dimension for each MIMF decreases with an increasing timescale, as seen in panel (a), the other two panels are not well elaborated.

**A3.** *We thank the Referee for this comment. $D_0$, $D_1$, and $D_2$ are strictly related to different properties of physical systems: (i) one purely geometric measure ($D_0$) providing us information on the coverage of the phase-space by the studied system's dynamics, (ii) one information measure ($D_1$)*

*giving us a measure of the information gained on the phase-space with a given accuracy , and (iii) one measure of correlations, i.e., mutual dependence, between phase-space points ($D_2$). Since the collective behavior of a system is given by physical processes operating at different scales, it is straightforward to look how they contribute to the topology of the phase-space, not only singularly (as in panel (a) of Figs. 5-6) but especially when considering all processes occurring below a selected scale (as in panel (b) of Figs. 5-6) and by looking separately at the atmosphere and ocean (as in panel (c) of Figs. 5-6).*

*We modified both the "Introduction" and Section 4.2 "Multiscale generalized fractal dimensions" to take care of this comment, introducing more details on the fractal dimensions in terms of their physical meaning (at least for $D_0$, $D_1$, and $D_2$), together with a more detailed description of Figures 5-6.*

**C4.** The authors cleverly described the experiments. To reproduce the work, one needs to understand all the mathematical formulas. In the scientific method, some time calculation and mathematical expression do not match as most of the calculation procedure follow fundamental statistical programming. It needs a concise explanation of calculating all these quantities like system attractors, phase space, and correlation dimensions. The description of these quantities introduced in the manuscript is very dubious and complex to replicate. The reviewer is thankful for providing data sets. But it becomes worthy if it includes an explanation of how to reanalyze these data sets.

**A4.** *We thank the Referee for this suggestion. Our full system consists of 36 variables, thus we are working on a 36-D space. For visual purposes, we reduced our 36-D space to a 3-D subspace by looking at the behavior of the three selected dynamical variables (i.e., $T_{o,2}$, $\Psi_{o,2}$, and $\psi_{a,1}$) in a 3D plot. This allows us to investigate the 3D projection of the full system phase-space attractor, i.e., the set of values toward which our system tends to evolve.*

*Concerning the calculation of the generalized fractal dimensions, we used the approach proposed by Hentschel and Procaccia (1983) consisting of partitioning the phase-space into hypercubes and then measuring the probability of finding a given hypercube filled by points and/or its generalization to a statistical order q (as also described at lines 40-52 of the submitted version).*

*This comment has been addressed by adding more information and details as Supplementary Material.*

**C5.** Finally, the reviewer appreciates the work of the authors. Still, it needs a bit more simplification and incorporating the issues mentioned above.

**A5.** *We really thank the Referee for his/her nice words on our manuscript and we have done our best to be more precise and more clear in the revised version of the manuscript.*

**Referee #2**

General comments:

The authors propose combining two apparently contradictory analysis techniques to the outputs of a low (36) dimensional dynamical ocean - atmosphere model. The first, makes a nontrivial decomposition of the 36 dimensional signal into series with well-defined time scales, the second analyses the phase spaces assuming the existence of scale invariant properties. The justification and interpretation of this is opaque.

While the authors question the utility of conventional analysis techniques, at least the latter can be interpreted in straightforward manners. The interpretation of their results is nontrivial.

*We thank the Referee for raising some points that can be helpful for improving the presentation and clarity of our findings. Most of all, we did not mean to question the utility of conventional analysis techniques, but rather acknowledge their intrinsic limitations and attempt to explore the potentials of a combination of two "non-conventional" techniques to provide additional information.*

*We also want to stress that the two methods should not be seen as "apparently contradictory" as emphasized by the reviewer. The modes extracted in the first analysis step have no well-defined time scales but are instead characterized by scales that are time-dependent. This is one of the main novelties of the Empirical Mode Decomposition and its multivariate extension we used here (i.e., the MEMD) as compared to fixed-scale decomposition methods like wavelets. The extracted modes can be seen as representative of fluctuations at a typical scale that is the average of the instantaneous scales derived from a given mode via the Hilbert Transform. Moreover, the second analysis step, i.e., the generalized fractal dimensions, requires to have scale invariant properties in the phase-space of a given system, thus working (essentially) on measuring the geometric properties of the system trajectory and information on how to reconstruct it by measuring the information dimension $D_1$ and q-tuplet correlations $D_{q>1}$. This means that there are no a priori constraints on understanding a system using $D_q$. Thus, the two methods are not contradictory but rather complementary.*

Detailed comments:

**C1.** The notation is not easy to follow. Please explain the curly bracket notation used throughout:

On the left, a bold symbol "s" is used which is standard for indicating a vector. Why do the authors (apparently needlessly) add curly brackets and then an explicit restriction as a subscript?
Further, there is the bizarre looking symbol that is also not adequately explained.

**A1.** *We thank the Referee for this suggestion. Indeed, we agree that the notation using curly brackets has been partly misleading, since the left-hand side of the equation was originally intended to represent a sequence of multivariate observation vectors, while the right-hand side was supposed*

*to clarify the structure of each of those vectors composed of k scalar properties, the latter of which however was lacking clarity in our notation. This aspect has been clarified in our revised manuscript by removing the additional sequence notation. Moreover, the "bizarre looking" symbol has been changed to $D_q$, following the notation used in Alberti et al. (Chaos, 2020). We modified Section 3.1 "Multivariate Empirical Mode Decomposition MEMD)" accordingly as well as we took care of modifying other parts through the manuscript.*

**C2.** When discussing the mathematical properties of the usual decompositions ("completeness, convergence, linearity, and stationarity") it is stated that "these conditions are not usually met when real-world geophysical data are analyzed". This is confusing since the mathematical properties of Fourier or other decompositions are valid irrespective of any application. I think the authors meant to question the appropriateness of such decompositions for their specific application? However, this is a mathematical question that cannot be answered without reference to a specific assumed mathematical framework. In the paper the authors do not analyze empirical data at all but rather model outputs. Contrary to real empirical series, their series are therefore taken from a well-defined mathematical framework given by dynamical systems theory. Please explain why standard decompositions are not adequate for studying such model outputs and why there is a need for them to be replaced by decompositions with quite nontrivial interpretations and properties.

*A2. We thank the Referee for this important suggestion. As also highlighted in our reply to Referee #1 we actually had to clarify the sentence on properties met by real-world data (note that our manuscript does not exclusively utilize low-order model output, but also reanalysis data, which in our opinion would qualify as "empirical") for which linearity and stationarity assumptions are often not met. Indeed, we fully agree that mathematical properties of the decomposition methods themselves are surely valid irrespective of any application. As also suggested by the Referee, we referred to the use of adaptive methods that can be justified to overcome some limitations of fixed-basis methods such as linearity and stationarity assumptions. Moreover, adaptive methods (as the MEMD) could be more suitable for reducing some mathematical assumptions and a priori constraints.*

*Although we use the MEMD on a well-defined framework derived from dynamical systems theory, the reduced a priori constraints and the limited number of intrinsic components that can be visually inspected could be an advantage with respect to standard decompositions. Another advantage concerns the combination with generalized fractal dimensions: if we, for example, use Fourier decomposition we would have a large number of (harmonic) oscillating components at different fixed frequencies that should be summed up for exploiting our proposed procedure. Furthermore, if we, for example, use wavelets we would deal with some a priori assumptions on the decomposition basis onto which we are projecting our data that could produce misleading results in our procedure of evaluating fractal measures on a priori fixed scales. Thus, we do not question the appropriateness of conventional analysis techniques, but rather acknowledge that they (as well as any other) have intrinsic limitations in what we can learn from them. We modified accordingly Section 3 "Methods".*

**C3.** Also in the Methods section, it is stated that the authors "put forward a novel approach based on combining two different data analysis methods for Multivariate Empirical Mode Decomposition and generalized fractal dimensions". What is confusing is that while the MEMD analyzes time series in real space, in their application, the generalized fractal dimensions analysis is carried out in a quite different space - the phase space of each series. The result is that for each time series with characteristic time scale t, that the corresponding phase spaces are assumed to be scaling. In other words, while there are essentially no scaling properties in real space, it is assumed that there will be nontrivial scaling properties in the corresponding phase space. The approach is presumably justified if the characterizing these scaling properties via generalized fractal dimensions will help understand the system. At this point one wonders whether the conventional Fourier spectrum of each t scale series might have been easier to interpret, to understand. All this needs explanation, clarification.

*A3. We really appreciate this comment since it allows us to better underline our main aim. We have been interested in investigating how phase-space properties (geometry, correlations) change when dynamical components at different mean scales with different dynamics are considered. In other words, we have been interested in looking at the role of scale-dependent phenomena in defining the whole properties of a system. Global measures proposed in the past only allow us to investigate the statistical, topological, or geometric scaling properties of the whole system; conversely, our proposed approach allows us to investigate how the different scales contribute to the global properties of a system. Moreover, our framework also provides consistency with established measures for characterizing time series from a global (not scale-resolved) perspective, since the scale-dependent measures we evaluate converge to the associated global measures as all scales are considered, i.e., when the full system dynamics, composed by all accessible scales, is reached, Within this framework, our approach could be promising for investigating scale-dependent properties, as measured by fractal dimensions, of the system. We are indeed interested in nonlinear variability characteristics at different time scales, thus employing for example Fourier decomposition would leave us with perfectly linear and stationary harmonic functions as components, which do not carry any information on nonlinear dynamics, unless when studying their mutual phase relationships, leaving out the high-order statistical properties and only focusing on the autocorrelation function (i.e., the second-order moment). Otherwise, by looking at the behavior of fractal dimensions we can explore how the different scales contribute to change the phase-space properties that cannot be highlighted by using the conventional Fourier spectrum.*

*We have added more explanations and clarifications on those aspects at the end of Section 3.2 "Multivariate and multiscale generalized fractal dimensions" (lines 220-234).*

*C4.* In particular, when the generalized fractal dimensions are estimated, the authors need to show that there are indeed some phase space scaling properties. Using mathematical definitions such as eqs. 1-3 - where the small scale limits are taken - has only a formal validity when the definitions are applied to numerical model outputs, especially when the latter has been subjected to cubic spline interpolation which makes the small scales artificially smooth. In practice, one needs to display scaling behaviour over at least an order of magnitude or so in scale in order for any fractal dimension estimates to be convincing. The authors must therefore display some of their scaling plots - not just logarithmic slopes that have already been interpreted in terms of dimensions.

*A4. We agree with this comment. To be clearer and more convincing, we display in Figs. 1 and 2 of this response letter the scaling behavior for the correlation integral for the two cases C=0.008 and C=0.015 at different timescales. We choose to show here only the correlation integral since it can be faster evaluated than other moments (cfr. Grassberger and Procaccia, 1983). We show here that there exists at least an order of magnitude in scale over which a scaling behavior is observed. A similar behavior is also observed when considering the reanalysis data as shown in Fig. 3 of this response letter for the different regions. Taking also into consideration a corresponding comment by Referee #1, we added Supplementary Material with more details on the computation of fractal dimensions and scaling plots to the revised version of our manuscript.*

*We would further like to remark that the cubic spline interpolation does not produce artificially smoothed small scales since it does not act on the data themselves but only on local extreme values of the data to extract intrinsic oscillating components from the data. Thus, the shape of the raw data is not changed and generally the (M)EMD extracts scale-dependent components that are smoother as the largest scales are approached. We also fixed this point in Section 3.1 "Multivariate Empirical Mode Decomposition (MEMD)".*

[Figure]

**Fig. 1** The log-log scaling plots of the correlation integral C(r) as a function of r (normalized with the respect to the largest possible separation between points in the phase-space represented by $r_0$) at different scales represented by colors for the case C=0.008. The lines refer to the power law fit in the limit r→0.

[Figure]

**Fig. 2** The log-log scaling plots of the correlation integral C(r) as a function r (normalized with the respect to the largest possible separation between points in the phase-space represented by $r_0$) at different scales represented by colors for the case C=0.015.
The lines refer to the power law fit in the limit r→0.

[Figure]

**Fig. 3** The log-log scaling plots of the correlation integral C(r) as a function r (normalized with the respect to the largest possible separation between points in the phase-space represented by $r_0$) at different scales represented by colors for the reanalysis data.
The lines refer to the power law fit in the limit r→0.

**C5.** In this regard, I could also add that figs. 9 and 11 are almost certainly largely spurious. This is because typically for moments of order q≈>3-4, the moments are completely dominated by a single hypercube (a "second order multifractal phase transition") so that for larger q, the values will depend sensitively on the exact details of the input series. Similarly for q<0 most if not all the values will likely be spurious essentially due to the statistics of the very sparsely populated regions of phase space (the very low probability regions, see e.g. the discussion in ch. 5 of [Lovejoy and Schertzer, 2013]). In other words over most of the range of moments given in the figure (-20<q<20), the dimensions are likely to be spurious.

*A5. We thank the Referee for raising this important point on the statistical significance of higher-order moments. We are aware that this is a crucial point, especially when working with scale invariant features measured via structure functions, detrended fluctuation analysis, and spectral methods (as for wavelets). To deal with this problem and to support the statistical significance of our results we have followed the approach described in Ch. 5 of Lovejoy and Schertzer (2013) to evaluate the maximum moments as those derived from the tail of the cumulative distribution function of the data. Since we deal with the investigation of scale-dependent fractal dimensions, we evaluate the cumulative statistics at different scales. As shown in Figs. 4-6 in this response letter, we observe that extreme fluctuations follow a power law decay leading to the divergence of the 6-th order moment and the 4-th order moment for C=0.008 and C=0.015, respectively. Thus we have now fixed our range of moments -6<q<6 and -4<q<4 for C=0.008 and C=0.015, respectively, and modified Figs. 9-11 accordingly, without changing the previously described results qualitatively. Similar results have also been obtained for the reanalysis data (see Figs. 7-8 in this response letter), thus we fix here our range of moments to -3<q<3.*

*We also inserted Figs. 4-8 of this response letter as parts of our new Supplementary Material (see Figs. S6-S10).*

[Figure]

**Fig. 4** Complementary cumulative distribution functions at different scales as reported by different colors for the case C = 0.008. The lines refer to the power law fit of the tail.

[Figure]

**Fig. 5** Same as in Fig. 4, but for C = 0.015.

[Figure]

**Fig. 6** The power law scaling exponent $q_D$ as a function of the different scales for the case C = 0.008 (black asterisks) and C = 0.015 (red diamonds). The minimum $q_D$ has been chosen to set the range of statistically significant moments.

[Figure]

**Fig. 7** Same as in Fig. 4, but for the reanalysis data.

[Figure]

**Fig. 8** Same as in Fig. 6, but for the reanalysis data.

**C6.** Finally, the interpretation of the key figures 5-8 is not at all obvious. Calling these characterizations "topological, geometric" is unhelpful and/or misleading since they are actually statistical exponents without any straightforward relationship to the phenomenon under study. The authors could note that whereas a white noise signal would give a correlation dimension equal to the dimension of the phase space itself (it is space filling), that a Brownian motion in a space $d \geq 2$ has a constant dimension = 2.

*A6. We thank the Referee for this comment. We have attempted to further improve the clarity of our manuscript during the revision, especially in the paragraphs introducing some key concepts and/or describing key features. We are referring to topological and geometric since some measures are able to give us information on phase-space properties. For example, $D_0$ is a measure of the filling of the phase-space, thus providing a measure on the coverage of the phase-space by the studied system's dynamics, $D_1$ provides a measure of the information gained on the phase-space with a given accuracy , and the $D_{q>1}$ provide measures of q-tuplet correlations, i.e., mutual dependence, between phase-space points. This explains why we used the terms topological and geometric in our manuscript. We modified accordingly our "Introduction" as well as we give a deeper description of our key figures 5-8 (see Section 4.2 "Multiscale generalized fractal dimensions", lines 307-352).*

Reference:

Lovejoy, S., and Schertzer, D., The Weather and Climate: Emergent Laws and Multifractal Cascades, 496 pp., Cambridge University Press, 2013.

*Thanks a lot for this reference that we have also cited in the revised version of our manuscript.*

---

## Referee Report (RR1)

**More comments on "Multiscale fractal dimension analysis of a reduced order model of coupled ocean-atmosphere dynamics"**

Tommaso Alberti, Reik V. Donner, and Stéphane Vannitsem

General comments:

The manuscript has been improved, in particular, it is more readable, easier to follow although it is still very technical.

The authors are not alone in this tendency to develop more and more sophisticated algorithms that yield results further and further removed from the original physical problem. These methods are not wrong, the problem is more with the interpretation of the results. Recall that even the (old) Fourier technique was sufficiently difficult to interpret that it led to the "missing quadrillion" in atmospheric variability that was only recently discovered (2015) and that is still widely ignored! Therefore my point for discussion (below) is somewhat optional (I think it could potentially better situate the authors' technique), but not essential for publication In other words, if the authors can respond to the minor comments below, then the paper could be published).

Discussion point:

My main issues are still associated with the rather indirect and difficult to interpret method that is introduced. For example, a key empirical feature of macroweather temperatures is their temporal scaling over wide ranges (typically ≈ 1 month up to decades and longer) that involves long range system memory. It has recently been shown that such memories arise as classical consequences of the classical heat equation when the correct radiative-conductive boundary conditions are used [*Lovejoy et al.*, 2021], [*Lovejoy*, 2021]. Both the empirical finding itself (that can be used for example for monthly, seasonal forecasting, land and ocean, [*Del Rio Amador and Lovejoy*, 2021a; *Del Rio Amador and Lovejoy*, 2021b]) and the rather general (heat storage) mechanism (that applies to both land and ocean), bring into question the strong assertion (line 26) that "low-frequency variability (LFV) is strictly related to the ocean.".

Rather than investigating the scaling in a rather abstract phase space constructed with a complex sifting procedure, shouldn't we first attempt to understand the rather fundamental real space scaling that has still not been satisfactorily explained by dynamical systems theory?

Minor comments:

1 Line 38: box-counting was proposed in the 1950's, not by Ott 2002.

2. Line 50 and several other places: the scaling exponents $D_q$ characterized the statistics of the phase space scaling; calling them "geometric" is anachronistic (from Mandelbrot) and misleading. Elsewhere $D_q$ is even attributed "topological properties" even though the phase space is considered to be a set of isolated points (i.e. with topological dimension zero – or after interpolation, topological dimension=1). It is the phase space density of points whose density statistics are characterized by $D_q$.

3. Line 50: there is a conceptual slippage. It is stated (blue):
with $\Theta(\cdots)$ being the Heaviside function. More specifically, $D_0$ is a purely geometric measure providing us information on the coverage of the phase-space by the studied system's dynamics, $D_1$ is an information measure giving us a measure of the information gained on the phase-space with a given accuracy, while $D_2$ is a measure of correlations, i.e., mutual dependence.

However, the $D_q$ are exponents characterizing the rate at which the sparseness ($D_0$), the information ($D_1$), the correlations ($D_2$) *change with scale* – i.e. NOT the values at any given scale. There is then confusion because the next line: "without exploring how these properties evolve at different scales" refers now to scales in real space rather in phase space.

4. Line 110, one discusses scale invariant features over a wide range of scales and then refers to a recent review (Franzke et al 2020). On the one hand, it would be of interest to see if the model has realistic real space scaling

properties, and the slightly older monograph [*Lovejoy and Schertzer*, 2013] covers far more relevant material since it includes spatial scaling (the main source of temporal scaling) as well as the shorter (weather) time scales covered by the authors' model.

5. Line 123: The authors mention: "nonlinearity and non-stationarity properties of signals". We should be clear that signals are simply signals, they are neither nonlinear nor nonstationary. The latter are properties of processes or of models or of infinite ensembles – i.e. of theoretical constructs. In other words, the pertinence (or otherwise) of MEMD must be justified (or not) by the theoretical framework from which the signal is assumed to issue. Therefore the argument should be based on the characteristics of the 36 component dynamical system that is assumed to be a good model of the real world system.

6. Eq. 11, the original exponent (q) was correct!

7. Line 369: the effect of sample size and its implications for spurious scaling may be due either to first order multifractal phase transitions (from the probability tail as indicated here), or from second order phase transitions (see ch. 5, section 5.3, [*Lovejoy and Schertzer*, 2013].

8. Line 380: The "multifractal width" is in fact an ad hoc way of quantifying multifractality. It is not optimal since it is generally not a characteristic of the process, since it is sensitive to the sample size (this is due to multifractal phase transitions either the first order transitions mentioned on line 369 or to second order transitions c.f. above reference). That is why a better alternative is simply to use the co-dimension of the mean (= $d-D_1$ where d is the dimension of the phase space).

9. Although there is much discussion about scaling properties in phase space, there is no mention of the fundamentally important scaling properties in real space.. It would be valuable if the authors could discuss how their results help us understand (or not), this basic feature of temperature and other fields.

References:

Del Rio Amador, L., and Lovejoy, S., Using regional scaling for temperature forecasts with the Stochastic Seasonal to Interannual Prediction System (StocSIPS), *Clim. Dyn.*, *in press* doi: doi: 10.21203/rs.3.rs-326161/v1, 2021a.
Del Rio Amador, L., and Lovejoy, S., Long-range Forecasting as a Past Value Problem: Untangling Correlations and Causality with scaling, *Geophys. Res. Lett.*, *under review*, 2021b.
Lovejoy, S., The Half-order Energy Balance Equation, Part 1: The homogeneous HEBE and long memories, *Earth Syst .Dyn.* , *(in press)* doi: https://doi.org/10.5194/esd-2020-12, 2021.
Lovejoy, S., and Schertzer, D., *The Weather and Climate: Emergent Laws and Multifractal Cascades*, 496 pp., Cambridge University Press, 2013.
Lovejoy, S., Procyk, R., Hébert, R., and del Rio Amador, L., The Fractional Energy Balance Equation, *Quart. J. Roy. Met. Soc.* , 1–25 doi: https://doi.org/10.1002/qj.4005, 2021.

---

## Author Response (AR2)

Dear Editor,

first of all, we would like to express our sincere thanks for your overall positive evaluation of our manuscript and for your particular thoughts on our approach and your encouragement in staying rigorous and formal to keep the scientific level up as high as we can. With this letter, we are submitting a revised version of our manuscript entitled "Multiscale fractal dimension analysis of a reduced order model of coupled ocean-atmosphere dynamics". We carefully considered and addressed all minor comments and suggestions raised by Referee #2 to improve our manuscript.

We are confident that the revised version allows us to present our results in a more detailed and appropriate way. In the following, we provide a point-by-point reply (in italics) to all comments (in normal font) of Referee #2.

Sincerely,
Tommaso Alberti, Reik Donner, and Stéphane Vannitsem

**Referee #2**

General comments:
The manuscript has been improved, in particular, it is more readable, easier to follow although it is still very technical.
The authors are not alone in this tendency to develop more and more sophisticated algorithms that yield results further and further removed from the original physical problem. These methods are not wrong, the problem is more with the interpretation of the results. Recall that even the (old) Fourier technique was sufficiently difficult to interpret that it led to the "missing quadrillion" in atmospheric variability that was only recently discovered (2015) and that is still widely ignored! Therefore my point for discussion (below) is somewhat optional (I think it could potentially better situate the authors' technique), but not essential for publication In other words, if the authors can respond to the minor comments below, then the paper could be published).

*We thank the Reviewer for his/her positive evaluation of our revised manuscript. We think that this general comment could really open a wide discussion on the topic of data analysis in general. While we agree that sophisticated algorithms could lead to additional difficulties in interpreting the associated results, it is also surely true that this also applies to less sophisticated ones. Indeed, we could easily question on the reliability of describing a plethora of natural phenomena, which are nonlinear and/or non-stationary (as also the Reviewer stated in their minor comments below), by using stationary methods or via fixed-basis decomposition methods that could not be representative of the phenomenon under study. We think that completely answering or solving this debate is really difficult and likely not possible within just one paper. However, we considered the Reviewer's suggestions to improve the clarity of our manuscript.*

Discussion point:
My main issues are still associated with the rather indirect and difficult to interpret method that is introduced. For example, a key empirical feature of macroweather temperatures is their temporal scaling over wide ranges (typically ≈ 1 month up to decades and longer) that involves long range system memory. It has recently been shown that such memories arise as classical consequences of the classical heat equation when the correct radiativeconductive boundary conditions are used [*Lovejoy et al.*, 2021], [*Lovejoy*, 2021]. Both the empirical finding itself (that can be used for example for monthly, seasonal forecasting, land and ocean, [*Del Rio Amador and Lovejoy*, 2021a; *Del Rio Amador and Lovejoy*, 2021b]) and the rather general (heat storage) mechanism (that applies to both land and ocean), bring into question the strong assertion (line 26) that "low- frequency variability (LFV) is strictly related to the ocean.".

Rather than investigating the scaling in a rather abstract phase space constructed with a complex sifting procedure, shouldn't we first attempt to understand the rather fundamental real space scaling that has still not been satisfactorily explained by dynamical systems theory?

*We thank the Reviewer for this important point that surely needs to be further investigated in future work to obtain a complete understanding of real space scaling properties. However, our main aim is not to propose novel dynamical system models to explain energy/heat transfer or any other specific kind of physical process in the atmosphere-ocean coupled system. We rather propose and perform an investigation of the role of the different temporal scales in determining some of the key features of the studied model and how they can be reconciled with reanalysis data. The model we used here has been developed starting from the quasi-geostrophic equations describing the interaction between a two-layer atmosphere and a one-layer ocean over an infinitely deep quiescent ocean layer to investigate the ocean-atmosphere coupled dynamics. We are aware that different mechanisms could be responsible for developing low-frequency variability and, for this reason, we have slightly modified the sentence on line 26 accordingly, also introducing the suggested additional recent bibliography.*

Minor comments:

1 Line 38: box-counting was proposed in the 1950's, not by Ott 2002.

*We agree and modified this statement accordingly.*

2. Line 50 and several other places: the scaling exponents Dq characterized the statistics of the phase space scaling; calling them "geometric" is anachronistic (from Mandelbrot) and misleading. Elsewhere Dq is even attributed "topological properties" even though the phase space is considered to be a set of isolated points (i.e. with topological dimension zero – or after interpolation, topological dimension=1). It is the phase space density of points whose density statistics are characterized by Dq.

*We agree and modified this statement accordingly.*

3. Line 50: there is a conceptual slippage. It is stated (blue):

However, the Dq are exponents characterizing the rate at which the sparseness (D0), the information (D1), the correlations (D2) *change with scale* – i.e. NOT the values at any given scale. There is then confusion because the next line: "without exploring how these properties evolve at different scales" refers now to scales in real space rather in phase space.

*The Reviewer is right since the term "scale" as used here refers once to the phase space and then to the real space. We revised the corresponding paragraph accordingly.*

4. Line 110, one discusses scale invariant features over a wide range of scales and then refers to a recent review (Franzke et al 2020). On the one hand, it would be of interest to see if the model has

realistic real space scaling properties, and the slightly older monograph [*Lovejoy and Schertzer*, 2013] covers far more relevant material since it includes spatial scaling (the main source of temporal scaling) as well as the shorter (weather) time scales covered by the authors' model.

*We thank the Reviewer for this comment. Unfortunately it is not possible to perform such an analysis at this point, since the used model does not involve a sufficiently wide range of spatial scales. However, in addition to Franzke et al. (2020) we also refer to Lovejoy and Schertzer (2013) in our revised manuscript.*

5. Line 123: The authors mention: "nonlinearity and non-stationarity properties of signals". We should be clear that signals are simply signals, they are neither nonlinear nor nonstationary. The latter are properties of processes or of models or of infinite ensembles – i.e. of theoretical constructs. In other words, the pertinence (or otherwise) of MEMD must be justified (or not) by the theoretical framework from which the signal is assumed to issue. Therefore the argument should be based on the characteristics of the 36 component dynamical system that is assumed to be a good model of the real world system.

*We thank the Reviewer for this comment. We agree that nonlinearity and non-stationarity are properties of phenomena manifesting into signals. We modified the text accordingly.*

6. Eq. 11, the original exponent (q) was correct!

*We changed this accordingly.*

7. Line 369: the effect of sample size and its implications for spurious scaling may be due either to first order multifractal phase transitions (from the probability tail as indicated here), or from second order phase transitions (see ch. 5, section 5.3, [*Lovejoy and Schertzer*, 2013].

*We changed this accordingly.*

8. Line 380: The "multifractal width" is in fact an ad hoc way of quantifying multifractality. It is not optimal since it is generally not a characteristic of the process, since it is sensitive to the sample size (this is due to multifractal phase transitions either the first order transitions mentioned on line 369 or to second order transitions c.f. above reference). That is why a better alternative is simply to use the co-dimension of the mean (= d-D1 where d is the dimension of the phase space).

*We thank the Reviewer for this comment. We used the multifractal width since it is a simple and relatively easy to interpret quantitative concept to evidence what is also reported in Figs. 7-8, i.e., the different values of $D_q$ for different q. We prefer to avoid introducing yet another measure like the co-dimension of the mean for the benefit of the reader and for the sake of simplicity. We however added a few details on this aspect in the corresponding part of the manuscript.*

9. Although there is much discussion about scaling properties in phase space, there is no mention of the fundamentally important scaling properties in real space. It would be valuable if the authors could discuss how their results help us understand (or not), this basic feature of temperature and other fields.

*We thank the Reviewer for this comment. We added a few details on this aspect in the conclusion part of the manuscript.*

References:

Del Rio Amador, L., and Lovejoy, S., Using regional scaling for temperature forecasts with the Stochastic Seasonal to Interannual Prediction System (StocSIPS), *Clim. Dyn.*, *in press* doi: doi: 10.21203/rs.3.rs-326161/v1, 2021a.

Del Rio Amador, L., and Lovejoy, S., Long-range Forecasting as a Past Value Problem: Untangling Correlations and Causality with scaling, *Geophys. Res. Lett.*, *under review*, 2021b.

Lovejoy, S., The Half-order Energy Balance Equation, Part 1: The homogeneous HEBE and long memories, *Earth Syst .Dyn.* , *(in press)* doi: https://doi.org/10.5194/esd-2020-12, 2021.

Lovejoy, S., and Schertzer, D., *The Weather and Climate: Emergent Laws and Multifractal Cascades*, 496 pp., Cambridge University Press, 2013.

Lovejoy, S., Procyk, R., Hébert, R., and del Rio Amador, L., The Fractional Energy Balance Equation, *Quart. J. Roy. Met. Soc.* , 1–25 doi: https://doi.org/10.1002/qj.4005, 2021.